# CONCISE AND ORGANIZED PERCEPTION FACILITATES LARGE LANGUAGE MODELS FOR DEDUCTIVE REASONING

## ABSTRACT

Exploiting large language models (LLMs) to tackle deductive reasoning has garnered growing attention. It still remains highly challenging to achieve satisfactory results in complex deductive problems, characterized by plenty of premises (i.e., facts or rules) entailing intricate relationships among entities and requiring multi-hop reasoning. One intuitive solution is to decompose the original task into smaller sub-tasks, and then chain the multiple casual reasoning steps together in a forward (e.g., Selection-Inference) or backward (e.g., LAMBADA) direction. However, these techniques inevitably necessitate a large number of overall stages, leading to computationally expensive operations and a higher possibility of making misleading steps. In addition to stage-by-stage decomposition, we draw inspiration from another aspect of human problem-solving. Humans tend to distill the most relevant information and organize their thoughts systematically (e.g., creating mind maps), which assists them in answering questions or drawing conclusions precisely and quickly. In light of this, we propose a novel reasoning approach named Concise and Organized Perception (COP). COP carefully analyzes the given statements to efficiently identify the most pertinent information while eliminating redundancy. It then prompts the LLMs in a more organized form that adapts to the model's inference process. By perceiving concise and organized proofs, the deductive reasoning abilities of LLMs can be better elicited, and the risk of acquiring errors caused by excessive reasoning stages is mitigated. Furthermore, our approach can be combined with the aforementioned ones to further boost their performance. Extensive experimental results on three popular deductive benchmarks (i.e., ProofWriter, PrOntoQA and PrOntoQA-OOD) show that COP significantly outperforms previous state-of-the-art methods.

## 1 INTRODUCTION

The field of large language models (LLMs) has witnessed significant progress in complex reasoning with the advent of Chain-of-thought (CoT) prompting (Wei et al., 2022) and a series of related works (Kojima et al., 2022; Zhou et al., 2023; Qiao et al., 2022). These breakthroughs have yielded remarkable achievements in various applications, including arithmetic, commonsense, symbolic reasoning, etc., and have sparked widespread enthusiasm within the community to continuously explore the immense potential of LLMs in tackling complex reasoning tasks. In this work, we focus on deductive reasoning, which is considered as one of the most rigorous forms of logical reasoning, built upon logical rules of inference, such as modus ponens (Johnson-Laird, 1999; Goel, 2007; Adler & Rips, 2008; Johnson-Laird et al., 2015). It begins with a set of given premises and employs logically valid arguments to draw necessary conclusions. In domains such as science, mathematics, medicine and law, deductive reasoning plays a crucial role as it enables the derivation of definite inferences (Metaxiotis et al., 2002; Goswami, 2010; McCarty, 2013). Moreover, it serves as an inherent guiding mechanism, implicitly involved in problem-solving steps within the aforementioned arithmetic or commonsense reasoning scenarios.

Recently, the emergence of new datasets (Clark et al., 2021; Tafjord et al., 2021; Saparov & He, 2023; Saparov et al., 2023) has greatly promoted research in this area. Samples of these datasets are constructed from synthetically generated data and provided as natural language sentences. They

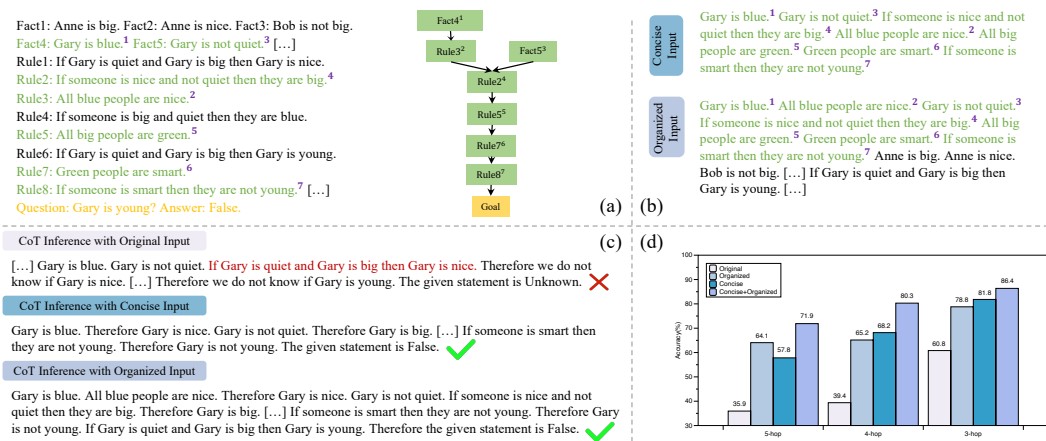

Figure 1: (a) A 5-hop example of ProofWriter dataset, showcasing the premises, question, and corresponding gold proof path for reference. Some facts and rules are omitted for brevity. (b) Corresponding reconstruction of concise and organized perception. Superscript serial numbers represent the logical order according to the gold proof. The concise input contains only relevant information but lacks organizational structure, while the organized input arranges statements in a consistent manner with the gold proof path, albeit including some redundant information. (c) LLMs output results. (d) Results of a confirmatory experiment.

describe the process of logical reasoning, primarily based on deduction rules. Figure 1(a) presents an example of ProofWriter (Tafjord et al., 2021), consisting of input premises (facts and rules) and a hypothesis whose truth is to be determined based solely on the in-context information. The complexity of the context directly affects the difficulty of deductive reasoning. When the attributes of entities and relations among entities become more intricate or the depth of the proof increases (i.e., multi-hop reasoning is required), the models face a higher risk of selecting the wrong step at some stage. This often leads to an incomplete proof and subsequently an incorrect answer. Figure 1(c) illustrates such a misleading step, where the original CoT incorrectly selects Rule1 (highlighted in red). This observation indicates that LLMs usually struggle with proof planning for more complex deductive reasoning problems, as thoroughly investigated in Saparov & He (2023).

One prevailing paradigm is to decompose complex problems into smaller sub-tasks (Creswell et al., 2023; Jung et al., 2022; Creswell & Shanahan, 2022; Kazemi et al., 2023; Wang et al., 2023), as LLMs tend to perform fairly well at single step inference, which helps compensate for their limitations in complex reasoning. It appears intuitive that these methods directly guide or constrain the planning process of LLMs. For example, Creswell et al. (2023) (SI) suggest alternating between selection and inference to generate a series of casual reasoning steps, while Kazemi et al. (2023) (LAMBADA) employ a more explicit manner to introduce backward chaining for high-level proof planning. Although they achieve considerable accuracy improvements, when confronted with more complex logical reasoning problems, it is still inevitable to engage in a substantial search for determining each step and still requires a relatively large number of overall steps. Consequently, this leads to computationally expensive operations and a higher possibility of making misleading or invalid steps.

In this paper, we draw inspiration from another perspective of human problem-solving. Rather than immediately searching for a solution when faced with a large amount of information and a question, humans are prone to streamline and organize the provided information in an orderly manner, such as constructing a mind map. This allows them to address the question more quickly and accurately by referring to the mind map. Drawing a parallel to LLMs, apart from the aforementioned methods from the perspective of *how to plan*, this insight inspires us to consider an alternative angle, which is reducing the difficulty of planning, or in other words, *easy to plan*.

We further obtain two sources of inspiration that offer practical implementation ideas. The first is the field of strategic rules, which mean the relevant rules supporting deduction to arrive at the intended conclusion (Jaakko & Sandu, 2006). To illustrate this, we can use the analogy of playing chess: the

vanilla rules determine whether one plays chess or something else whereas strategic rules determine whether one is a good or a bad chess player (Jaakko & Sandu, 2006). The same applies to deductive reasoning: to be an effective reasoner involves eliminating redundant or irrelevant information and making the relevant information more explicit. Secondly, as demonstrated in Saparov & He (2023), traversal direction affects reasoning. As the number of hops increases, the model becomes sensitive to the traversal direction of the ontology. This reveals that we need to organize the ordering of the context sentences in a progressively logical fashion that aligns with the model's inference process.

Taking the above factors into account, we propose a novel reasoning approach named Concise and Organized Perception (COP). Specifically, COP initially generates concept maps[1] which depict the hierarchical relationships among the given facts and rules, following deduction rules. This allows a comprehensive understanding of the input context. Next, based on the query that needs to be proved, COP identifies the most relevant information from the concept maps while eliminating redundancy, resulting in a mind map-like structure centered around the query node. After that, LLMs are prompted by the context sentences which are organized in a progressively ordered manner within one or more sequential sub-mind maps, in order to better adapt to the inference process of the model. We believe that such reconstruction perceives more concise and organized information, which noticeably reduces the difficulty of model inference and better elicits the deductive reasoning ability. Figure 1(b)(c) shows an example where LLMs are empowered to obtain the correct answer.

Meanwhile, we further conducted a simple confirmatory experiment by randomly selecting 196 samples and reconstructing the context based on the provided ground-truth proofs, as shown in Figure 1(b) [2]. The results in Figure 1(d) demonstrate that combining our approach with the CoT baseline yields a relative performance improvement of over 100% (35.9% vs 71.9%) in a 5-hop setting. The results also indicate the complementarity between concise and organized perception. Furthermore, since our approach naturally comes from a different perspective, it can be seamlessly combined with other popular methods, such as SI or LAMBADA, to further enhance their performance.

In summary, we make the following contributions:

1. We introduce the Concise and Organized Perception (COP) approach, which significantly reduces the difficulty of LLMs proof planning (i.e., easy to plan), and better elicits their deductive reasoning abilities.

2. "Concise" and "Organized" are both effective strategies, and simply combining them with vanilla CoT achieves excellent performance. Moreover, integrating our method with other approaches like SI or LAMBADA can provide additional benefits.

3. Extensive experimental results on three popular deductive benchmarks (i.e., ProofWriter, PrOntoQA and PrOntoQA-OOD) demonstrate that our method achieves state-of-the-art performance.

## 2 RELATED WORK

Large language models (LLMs) have demonstrated impressive few-shot learning capabilities (Brown et al., 2020; Raffel et al., 2020; Chung et al., 2022; Ouyang et al., 2022; Touvron et al., 2023). However, they often struggle when it comes to logical reasoning tasks (Rae et al., 2021). The deductive reasoning problem we discuss in this paper is one of the most rigorous and common types of logical reasoning (Johnson-Laird, 1999), and recent work has shown that LLMs, combined with in-context learning (ICL) and chain-of-thought (CoT) prompting, are capable of deductive reasoning to an extent (Huang & Chang, 2022; Qiao et al., 2022; Nye et al., 2021; Wei et al., 2022; Kojima et al., 2022; Lewkowycz et al., 2022).

Those works in adapting LLMs for logically deductive reasoning tasks can be broadly categorized into three groups: 1) approaches that aim to fine-tune LLMs in order to directly produce the final answer, keeping reasoning implicit (Clark et al., 2021; Lewkowycz et al., 2022); 2) approaches

---

[1]Concept maps are free-form diagrams representing relationships between concepts and ideas, helping people organize and structure knowledge.https://en.wikipedia.org/wiki/Concept_map.

[2]The detailed prompt is omitted, see Appendix A.1, and notice that this implementation is merely demonstrative, and differs from the actual method as no ground-truth can be utilized, see Section 3 and Figure 2 for details.

that encourage LLMs to explicitly generate reasoning steps, but all of them are produced in a single stage (Cobbe et al., 2021; Dalvi et al., 2021; Zelikman et al., 2022; Wei et al., 2022; Kojima et al., 2022); and 3) approaches that utilize LLMs to generate each reasoning step one at a time. Jung et al. (2022) regard the output of each stage as a separate new question while Zhou et al. (2023) break the problem down into simpler components that can be solved individually. Selection-Inference (Creswell et al., 2023) alternates between selection and inference to generate a series of casual reasoning steps, and LAMBADA (Kazemi et al., 2023) develops a backward chaining algorithm to decompose reasoning into sub-modules. The latter is currently the prevailing paradigm, and as we mentioned above, our method is complementary to these existing approaches.

Our approach is inspired in part by the following work (Creswell et al., 2023; Saparov & He, 2023). Creswell et al. (2023) carried out a comprehensive evaluation of LLMs on 50 tasks that probe different aspects of logical reasoning. They observed that the performance of vanilla language models tends to decrease when they get presented with irrelevant facts alongside the ones relevant for reasoning, when they have to infer the relevant facts from memory, and as the questions start to require more steps of reasoning. These findings motivate us to focus on identifying the most relevant information and eliminating redundancy in our approach. In another study, Saparov & He (2023) investigated how reasoning ability is affected by the traversal direction of the ontology. They discovered that traversal direction affects reasoning, which prompted us to organize the ordering of the context sentences in a progressively logical fashion that adapts to the model's inference process.

Several recent works, such as LOGIC-LM (Pan et al., 2023) and Scallop (Zhang et al., 2023), integrate LLMs with symbolic reasoning to improve logical problem-solving. LOGIC-LM first utilizes LLMs to translate a natural language problem into a symbolic formulation. Afterward, a deterministic symbolic solver performs inference on the formulated problem. Scallop is similar to LOGIC-LM, but with probabilistic reasoning engine. However, this complex conversion for LLMs poses challenges and limits their performance. In contrast, we leave the logical rules (i.e., conjunction and disjunction) to be handled by LLMs in the final reasoning stage, therefore greatly increases the accuracy of generating the simplified representations of rules and facts. Another work, MindMap (Wen et al., 2023), introduces knowledge graph (KG) prompting that endows LLMs with the capability of comprehending KG inputs and facilitates LLMs to infer with a combined implicit knowledge and the retrieved external knowledge. Unlike our focus, MindMap primarily aims to enhance reasoning ability based on the knowledge graph. On the other hand, ToT (Yao et al., 2023) and GoT (Besta et al., 2023) generalize over the CoT from tree-like or graph-like structures. These methods allow LLMs to make decisions by considering multiple different reasoning paths. On the contrary, our method concentrates on reducing the difficulty of proof plan by reconstructing in-context information.

## 3 APPROACH

We present the Concise and Organized Perception (COP) reasoning approach, aiming to leverage the semantic comprehension and reasoning ability of LLMs to address complex deductive reasoning problems. Given a reasoning context consists of multiple deductive rules $\mathcal{R} = \{r_1, r_2, ..., r_n\}$ and facts $\mathcal{F} = \{f_1, f_2, ..., f_n\}$, which may include relevant, irrelevant, or misleading information, the task is to determine the veracity of the answer to a question $\mathcal{Q}$.

As illustrated in Figure 2, the proposed COP initially creates concept maps that highlight the hierarchical relationships among the provided rules and facts to obtain a comprehensive understanding of the problem context. Next, based on the provided question, COP identifies relevant contexts on the concept maps and generates a mind map-like structure centered around the query node. Subsequently, owing to the progressively organized manner of the mind map, COP creates a more concise and organized reasoning context which can be easily adapted to the inference process of LLM models. The details of these steps will be described in the following subsections.

### 3.1 GENERATION OF CONCEPT MAPS

It is generally not a wise strategy to hastily answer questions without fully grasping the entire context when performing reasoning tasks; otherwise, it easily leads to inaccurate or incomplete reasoning. Therefore, instead of starting with looking for relevant clues based on local information as previous methods (e.g., SI and LAMBADA) do, the first step of the proposed COP is to generate concept maps

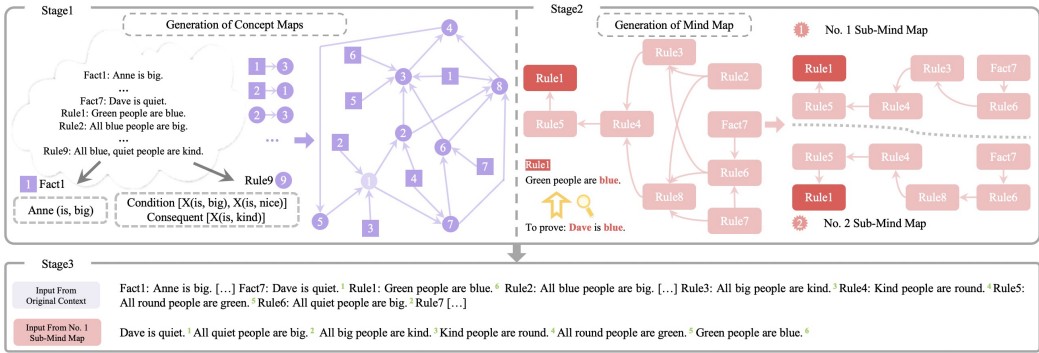

Figure 2: Overview of the proposed COP.

to obtain a comprehensive and structural understanding of the reasoning context, thus enabling the context reconstruction in the subsequent steps.

As mentioned previously, deductive reasoning problems involve a set of rules and facts. Each rule (i.e., a hypothetical proposition) describes an antecedent and a consequent which can be expressed as "*If antecedent then consequent*". The antecedent contains either a single condition or a combination of multiple conditions, with either conjunction (i.e., and) or disjunction (i.e., or) connecting each other in logic.

Imitating the process of human beings organizing thoughts, concept maps can be generated by leveraging directed edges to connect each rule to facts as well as rules whose consequents satisfy one or more of the conditions specified in the current rule.

**Simplified Representations of Rules and Facts.** The rules and facts are expressed in a variety of linguistic patterns (e.g., *all smart things are furry*, *if something is smart then it is furry*, *smart people are furry*), which makes it challenging to determine if the conditions of rules can be fulfilled by other rules or facts. In viewing of that, we take advantage of the strong information extracting ability of LLMs, with few-shot prompts, to create a unified and simplified representation for the facts and the rules presented in various language patterns as shown in Figure 2.

Given a fact "*The dog likes the cat*", it is changed into "*dog(like, cat)*", where "*dog*", "*like*" and "*cat*" are the subject, predicate and object respectively. As for rules such as "*If someone likes the cat and the dog chases the cat then it eats the mouse*", its conditions and consequents are transformed into "*[X(like, cat), dog(chases, cat)]*" and "*[X(eat, mouse)]*" respectively, where "*X*" can be substituted by any entities.

Notably, previous methods such as LogicLM utilize LLMs to convert the facts and rules into first-order logic languages then perform reasoning with a symbolic solver rather than LLMs. However, it is a great challenge to accurately translate a problem statement into a valid logical format using LLMs. On the contrary, we leave the logical rules (i.e., conjunction and disjunction) to be handled by LLMs in the final reasoning stage, therefore greatly increases the accuracy of generating the simplified representations of rules and facts.

**Connecting of Rules and Facts.** With the simplified representations of rules and facts, it naturally becomes easy to pair facts and rules. Facts like "*dog(like, cat)*" can be connected to rules with a condition "*X(like, cat)*" or "*dog(like, cat)*". Rules with consequent such as "*[X(eat, mouse)]*" can be connected to rules with a condition "*X(eat, mouse)*" or "*cat(eat, mouse)*". In this way, facts and rules are connected with directed edges, eventually forming one or more concept maps, as there might be isolated rules or facts, representing the structural understanding of the entire context.

## 3.2 GENERATION OF MIND MAP

In section 3.1, concept maps representing the structural understanding of the entire reasoning context are generated. Once a query (the target fact to be proved/disproved) is given, we identify relevant clues from the concept maps to create a mind map with the question node at its center.

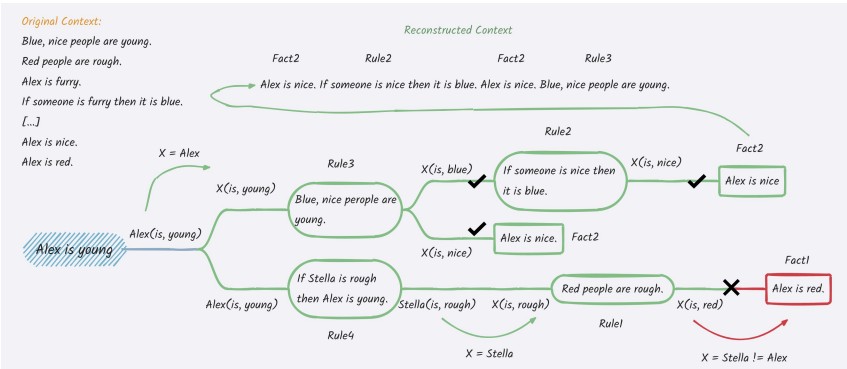

Figure 3: An example of sub-mind map pruning and context reconstruction.

**Simplified Representation of the given question.** Similar to the simplifying process of facts and rules in section 3.1, LLMs are employed to create simplified representations of the given question. For example, "*The tiger does not eat the rabbit*" is transformed as "*tiger(not eat, rabbit)*". If there are multiple statements in the question, "*Max is kind and a gorpus*" and "*Wren is brown or a tumpus*" for instance, they are simplified as "*[Max(is, kind), Max(is, gorpus)]*" and "*[Wren(is brown), Wren(is, tumpus)]*" respectively. Notably, we ignore the logical relationship contained in the question (i.e. conjunction and disjunction) which keeps consistent with the reconstruction of the rules. We leave that to be handled by LLMs in the final reasoning step. Moreover, to increase the accuracy of identifying relevant clues, we also generate the contrary statements of the given question. Therefore, the final reconstruction result of the given question "*Max is kind and a gorpus*" is "*[Max(is, kind), Max(is not, kind), Max(is, gorpus), Max(is not, gorpus)]*". The above process is done in one inference call of LLMs with few-shot prompts.

**Generation of the mind map.** Then we are able to identify the relevant rules and facts with the simplified question using the same way as we connect rules and facts when constructing the concept maps. Once we find the relevant rules and facts, we perform a $\mathcal{D}$-depth searching starting from each of them on the concept maps, where $\mathcal{D}$ is the max reasoning depth. In this way, a mind map based on the given question is constructed.

## 3.3 CONTEXT RECONSTRUCTION

Figure 3 presents an example of a mind map. Given a question "*Alex is young*" and its mind map generated in previous steps, the question is relevant with Rule3 and Rule4, whose simplified consequents are "X(is, young)" and "Alex(is, young)" respectively. Therefore the mind map naturally consists of two sub-mind maps. Every sub-mind map consists of several logically related rules and facts with directed connections.

To determine whether the given question is true or not, a reasoning context for each possible sub-mind map should be constructed to prompt the reasoning of LLMs. Therefore, to reduce the cost of reasoning, we perform sub-mind map pruning before conducting the final reasoning.

**Sub-Mind map Pruning.** Due to the structural and connective nature of sub-mind maps, there are some strong priors we can utilize to prune the sub-mind maps. Figure 3 presents an example of sub-mind map pruning. Even though the logical relations between the conditions of a rule are ignored in the sub-mind map, it can be easily inferred that "X" in the conditions of Rule3 should be substituted by "Alex", otherwise it is impossible to prove whether the given question is true based on this sub-mind map. Similarly, it can be inferred that the "X" in the conditions of Rule1 should be substituted by 'Stella', yet the Fact1 connecting to Rule1 does not support that. Therefore Fact1 is removed from the sub-mind map. After that, none of the facts can be used with Rule1 to perform any reasoning, so it is also removed. By recursively repeating this process, the number of proposals of sub-mind maps can greatly reduces.

**Context Reconstruction.** After pruning the sub-mind maps, we reconstruct a reasoning context for each remaining sub-mind map. As illustrated in Figure 3, the rules and facts in the reconstructed

context are organized by traversing the sub-mind map from its leaf nodes to the root node which naturally adapts to the LLMs as we demonstrate in Figure 1, thus eliciting the deductive reasoning ability of LLMs. Moreover, compared with the original reasoning context, the reconstructed one has the advantage of being concise and greatly reduces the impact of misleading proofs. Subsequently, the reconstructed contexts are successively used to prompt the reasoning of LLMs until a true or false statement regarding the given question is made.

## 4 EXPERIMENTS

### 4.1 DATASETS

**(1) ProofWriter** is a commonly used logical reasoning dataset for testing LLMs' deductive reasoning ability. We used the open-world assumption (OWA) subset for testing. Each example in ProofWriter consists of four parts: known facts, known rules, target question, and label. The target question is the fact to be proved and the label is one of {PROVED, DISPROVED, UNKNOWN}. The dataset has five subsets, named $d5$, $d3$, $d2$, $d1$ and $d0$ respectively. $dx(x \in \{0, 1, 2, 3, 5\})$ part requires $\leq x$ hops for reasoning. We randomly sampled 600 examples in each part and ensured a balanced label distribution for testing.

**(2) PrOntoQA** is a synthetic logical reasoning dataset, in which each example is generated from a synthetic world model. We used the hardest fictional characters version of the dataset based on the open-source data generation[3]. Each example in PrOntoQA consists of three parts: known rules, target question, and label. The target question is the rule or the fact to be proved and the label is one of {TRUE, FALSE}. Similar to ProofWriter, PrOntoQA is divided into five parts depending on the depth of reasoning chains required, named $hop5$, $hop4$, $hop3$, $hop2$, and $hop1$ respectively. $hopx(x \in \{1, 2, 3, 4, 5\})$ part requires $x$ hops for reasoning. We randomly sampled 500 examples in each part and ensured a balanced label distribution for testing.

**(3) PrOntoQA-OOD** is another synthetic logical reasoning dataset, which contains different types of deduction rules. Similar to PrOntoQA, each example in PrOntoQA-OOD consists of three parts: known rules, target question and label. We used the generated data file *generated_ood_data.zip* based on the open-source repository[4] to construct a test set consisting of three types of deduction rules (i.e., AndIntro, AndElim and OrIntro). We randomly selected 100 samples for each type of rule from the original $hop2$ part.

### 4.2 EXPERIMENTAL RESULTS

#### 4.2.1 PERFORMANCE COMPARISON WITH STATE-OF-THE-ART METHODS

In this section, we perform a thorough comparison between our proposed method and the existing state-of-the-art methods (Standard Few-Shot, CoT (Wei et al., 2022), SI (Creswell et al., 2023) LOGIC-LM (Pan et al., 2023) and LAMBADA (Kazemi et al., 2023)) for deductive reasoning. Unless otherwise specified, all the experimental results of COP are based on GPT-3.5-Turbo Ouyang et al. (2022).

Table 1 shows the results on subsets of ProofWriter and PrOntoQA with various reasoning depths. The results of SI and LAMBADA are taken from (Kazemi et al., 2023). COP consistently achieves the highest label accuracy across all experimental settings. Notably, COP outperforms SOTA methods by a large margin on the the hardest Depth-5 subset of ProofWriter. It shows a remarkable 65.74% relative improvement compared to CoT and 23.15% compared to LAMBADA, which clearly demonstrates the effectiveness of COP.

Intuitively, the reasoning becomes more challenging as the depth of reasoning increases. (Saparov & He, 2023) demonstrated that the difficulty in proof planning might be an important cause why LLMs perform poorly on multi-hop reasoning tasks. We highlight that COP creates a concise and organized reasoning context which greatly simplifies the proof planning process, as a result of which notably enhances the reasoning accuracy of LLMs on multi-hop problems.

---

[3]https://github.com/asaparov/prontoqa/tree/v1
[4]https://github.com/asaparov/prontoqa

Table 1: Comparison of label accuracy on ProofWriter and PrOntoQA.

| Datasets/ | ProofWriter | | | | | | PrOntoQA | | | |
|---|---|---|---|---|---|---|---|---|---|---|
| Methods | d5 | d3 | d2 | d1 | d0 | average | 5-hop | 3-hop | 1-hop | average |
| Standard | 41.67 | 49.83 | 51.00 | 55.50 | 63.67 | 52.33 | 49.60 | 52.00 | 65.60 | 55.73 |
| CoT | 53.50 | 61.17 | 61.33 | 62.33 | 62.83 | 60.23 | 69.80 | 74.20 | 86.20 | 76.73 |
| SI | 46.00 | 51.00 | 56.00 | 61.00 | 97.00 | 62.2 | 45.00 | 52.00 | 97.00 | 64.67 |
| LogicLM | 70.11 | - | - | - | - | - | 93.20 | - | - | - |
| LAMBADA | 72.00 | 82.00 | 87.00 | 90.00 | 98.00 | 85.80 | 96.00 | 99.00 | 98.00 | 97.67 |
| **COP** | **88.67** | **90.67** | **91.43** | **92.50** | **98.50** | **91.72** | **99.20** | **99.60** | **100.00** | **99.60** |

### 4.2.2 PERFORMANCE COMPARISON ON DIFFERENT TYPES OF DEDUCTIVE RULES

To validate the performance on different deductive rules, we tested three deductive rules (i.e., AndIntro, AndElim, OrIntro) on PrOntoQA-OOD dataset as all the samples in ProofWriter and PrOnto belong to only one type of rule (i.e., modus ponens). For each of the three rules, we randomly selected 100 samples as the test set. The experimental results are reported in Table 2. Though LAMBADA archives relatively high performance on ProofWriter and ProntoQA, it

Table 2: Test on PrOntoQA-OOD.

| Methods | AndIntro | AndElim | OrIntro | Overall |
|---|---|---|---|---|
| Standard | 75.00 | 11.00 | 44.00 | 43.33 |
| CoT | **93.00** | **96.00** | 68.00 | 85.67 |
| LAMBADA | 23.12 | 57.21 | 34.50 | 38.33 |
| **COP** | 92.00 | 95.00 | **95.00** | **94.00** |

is limited to the modus ponens rule therefore losing efficacy to any other types of deductive rules. On the contrary, the proposed method is effective on various types of deductive rules (e.g., disjunction elimination). COP achieves comparable performance on AndIntro and AndElim, and significantly outperforms CoT on OrIntro with a relative improvement of 39.71%.

### 4.2.3 PROOF ACCURACY ANALYSIS

Previous studies have demonstrated the phenomenon that CoT predicts a correct label with incorrect reasoning chains (Saparov & He, 2023). To validate if it is the case for COP, we randomly selected 100 correctly answered samples from the Depth-5 setting of ProofWriter and manually checked the reasoning chain produced by LLMs with COP. According to our observation, only 7 out of 100 samples contain invalid reasoning steps, which indicates that the proposed COP does arouse the reasoning ability of LLMs and the experimental results reported above are faithful.

### 4.2.4 IS COP BENEFICIAL TO OTHER METHODS?

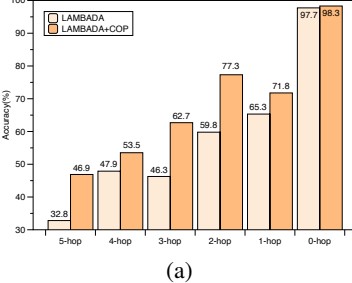
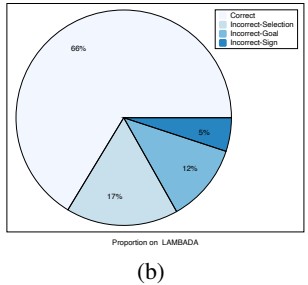
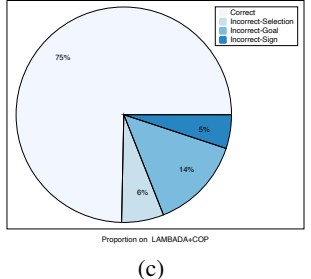

(a)            (b)            (c)

Figure 4: (a) Comparison of LAMBADA and LAMBADA+COP on Proofwriter. (b) The proportions of different error reasons of LAMBADA. (c) The proportions of different error reasons of LAMBADA + COP. Incorrect reasoning errors include selection errors, goal decomposition errors and sign agreement errors. And selection errors include fact check errors and rule selection errors.

The performance of LAMBADA and LAMBADA+COP on the ProofWriter $d_5$ subset with different inference depths are listed in Figure 4 (a). All the results are based on the LAMBADA code we reproduced and the base model of this experiment is GPT-3.5-turbo. Compared with the origi-

nal LAMBADA method, the performance of LAMBADA+COP under different inference depths is improved, proving the effectiveness of COP.

Figure 4(b)(c) show the proportion of correct reasoning and the proportion of different types of incorrect reasoning. We selected 100 test samples from the ProofWriter $d_5$ subset to manually check the error types of incorrect reasoning examples. As shown in the figure, equipped with COP, the proportion of selection errors (including fact check and rule selection modules in LAMBADA) drops significantly. The proportion of goal decomposition errors and sign agreement errors (goal decomposition and sign agreement modules are not affected by the context redundancy and disorder) are almost unchanged, which further proves that our COP can improve the success rate of other methods in the fact and rule selection steps.

### 4.2.5 PERFORMANCE WITH DIFFERENT LARGE LANGUAGE MODELS

The majority of experiments conducted in this paper were performed on ChatGPT-3.5-turbo(Ouyang et al., 2022). To study if the proposed COP is effective across different base models, we conducted experiments on text-davinci-003 with the Depth-5 set of ProofWriter. Table 3 illustrates a comparison between text-davinci-003 and gpt-3.5-turbo by reporting the label accuracy for samples with different reasoning hops. As shown in Table 3, COP consistently achieves high label accuracy using

Table 3: The performance comparisons using different LLMs.

| Base Models/ | text-davinci-003 | | | | | | GPT-3.5-turbo | | | | | |
|---|---|---|---|---|---|---|---|---|---|---|---|---|
| Methods | 5-hop | 4-hop | 3-hop | 2-hop | 1-hop | 0-hop | 5-hop | 4-hop | 3-hop | 2-hop | 1-hop | 0-hop |
| CoT | 34.38 | 49.30 | 56.71 | 45.36 | 46.77 | 51.98 | 45.31 | 54.93 | 65.67 | 48.45 | 47.58 | 58.19 |
| COP | 84.38 | 87.32 | 82.09 | 85.42 | 81.30 | 97.74 | **87.50** | **88.73** | **85.07** | **85.58** | **83.87** | **98.31** |

different LLMs, revealing its effectiveness across different LLMs.

### 4.2.6 NUMBER OF INFERENCE CALLS

In Figure 5, we compared the average number of inference calls per example under different reasoning depths. COP requires significantly fewer inference calls compared to LAMBADA, and the number of inference calls remains relatively stable as the number of hops increases, demonstrating our proposed COP's superiority in both effectiveness and efficiency.

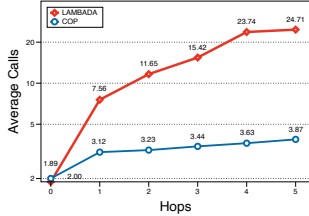

Figure 5: Comparison of inference calls.

## 5 CONCLUSION AND FUTURE WORK

In this study, we propose a reasoning approach called Concise and Organized Perception (COP) to effectively handle complex deductive reasoning problems, which serves as a valuable complement to previous stage-by-stage decomposition methods. By combining "Concise" and "Organized" strategies with vanilla CoT, we have achieved state-of-the-art performance on three popular deductive benchmarks. Besides, COP requires significantly fewer inference calls compared to decomposition-type methods (e.g., LAMBADA), highlighting our superiority in terms of both effectiveness and efficiency.

We believe our key insight on the proposal of easy-to-plan method has broader implications. However, when dealing with real-world scenarios, the measuring of retrieval ability and reasoning ability may confound. In addition, for more general reasoning tasks, the generation of a more appropriate concept or mind map-like structure requires further exploration. We plan to address these in future research.

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

## A PROMPTS

### A.1 PROMPTS USED FOR PROOFWRITER:

**For generating simplified representation of rules on ProofWriter:**

You are given some known rules. Extract the conditions and consequents of each rule and output follow the format of the given examples:

Examples:

Rules:

If someone sees the cat and they are not green then they see the cow. If the rabbit is kind and the rabbit sees the squirrel then the squirrel needs the rabbit. Rough people are cold. If someone sees the rabbit then they are not round. If someone sees the squirrel and they are not green then they need the squirrel. If someone eats the cow then they see the rabbit. Cold things are rough. If someone is cold then they eat the cow. Kind, rough people are round.

Output:

{"Rule1": {"conditions": ["X(see, cat)", "X(is not, green)"], "consequents": ["X(see, cow)"]},

"Rule2": {"conditions": ["rabbit(is, kind)", "rabbit(see, squirrel)"], "consequents": ["squirrel(need, rabbit)"]}, [...]

"Rule9": {"conditions": ["X(is, kind)", "X(is, rough)"], "consequent": ["X(is, round)"]}}

Rules:

If something visits the mouse and the mouse visits the dog then it is cold. If mouse likes the cat then it visits the dog. If something is cold then it likes the cat. If something is green then it sees the dog. If something likes the mouse then it sees the cat. If dog is green and cold then it likes the cat. If something is big and it visits the bear then the bear is green. Round things are rough.

Output:

{"Rule1": {"conditions": ["X(visit, mouse)", "mouse(visit, dog)"], "consequents": ["X(is, cold)"]},

"Rule2": {"conditions": ["mouse(like, cat)"], "consequents": ["X(visit, dog)"]},

"Rule8": {"conditions": ["X(is, round)"], "consequents": ["X(is, rough)"]}}

Rules:

{{The Rule to be simplified}}

Output:

**For generating simplified representation of facts on ProofWriter:** You are given some known facts. Output the facts following the format of the given examples:

Examples:

Facts:

The bear is green. The bear likes the cat. The bear likes the dog. The bear visits the dog. The cat is young. The cat does not see the bear. The cat sees the dog. The cat visits the bear. The dog is round. The mouse is not big. The mouse is cold.

Output: "Fact1": ["bear(is, green)"], "Fact2": ["bear(like, cat)"], "Fact3": ["bear(like, dog)"], "Fact4": ["bear(visit, dog)"], "Fact5": ["cat(is, young)"], "Fact6": ["cat(not see, bear)"], "Fact7": ["cat(see, dog)"], "Fact8": ["cat(visit, bear)"], "Fact9": ["dog(is, round)"], "Fact10": ["mouse(is not, big)"], "Fact11": ["mouse(is, cold)"]

Facts: {{The facts to be simplified}}

**For generating simplified representation of questions on ProofWriter:** You are given some known facts. Output the facts following the format of the given examples:

Examples:

Facts:

The bear is green.

Output:

["bear(is, green)", "bear(is not, green)"]

Facts:

The bear likes the cat.

Output:

["bear(like, cat)", "bear(not like, cat)]

Facts:

The cat does not see the bear.

Output:

["cat(not see, bear)", "cat(see, bear)"]

Facts:

The mouse is not big.

Output:

["mouse(is not, big)", "mouse(is, big)"]

Facts: {{Question to be simplified}}

**For reasoning on ProofWriter:**

Given a context consists of some known facts and rules, think step by step is the given statement true or false or unknown.

Examples:

Context:

The mouse is cold. If something is cold then it likes the cat. The mouse eats the dog. If something likes the cat and it eats the dog then it likes the rabbit. If something likes the rabbit then it visits the dog.,

Statement:

The mouse visits the dog.

Output:

The mouse is cold. If something is cold then it likes the cat. Therefore the mouse likes the cat. The mouse eats the dog. If something likes the cat and it eats the dog then it likes the rabbit. Therefore the mouse likes the rabbit. If something likes the rabbit then it visits the dog. Therefore the mouse visits the dog. Therefore the given statement 'The mouse visits the dog' is True.

Context:

The cat visits the dog. The mouse likes the rabbit. If something likes the rabbit and the cat visits the dog then it likes the dog. The mouse eats the dog. If something likes the dog and it eats the dog then it likes the rabbit. If something likes the 'rabbit then it visits the dog.

Statement:

The mouse does not visit the dog.

Output:

The cat visit the dog. The mouse likes the rabbit. If something likes the rabbit and the cat visit the dog then it likes the dog. Therefore the mouse likes the dog. The mouse eats the dog. If something likes the dog and it eats the dog then it likes the rabbit. Therefore the mouse likes the rabbit. If something likes the rabbit then it visits the dog. Therefore the mouse visits the dog. Therefore the given statement 'The mouse does not visit' is False.

... Context:

{{The context used to perform reasoning}}

Statement:

{{The question to be proved}}

Output:

## A.2 Prompts Used for PrOntoQA:

**For generating simplified representation of rules on PrONtoQA:**

You are given some known rules. Extract the conditions and consequents of each rule and output follow the format of the given examples:

Examples:

Rules:

Jompuses are not shy. Jompuses are yumpuses. Each yumpus is aggressive. Each yumpus is a dumpus. Dumpuses are not wooden. Dumpuses are wumpuses. Wumpuses are red. Every wumpus is an impus. Each impus is opaque. Impuses are tumpuses. Numpuses are sour. Tumpuses are not sour. Tumpuses are vumpuses. Vumpuses are earthy. Every vumpus is a zumpus. Zumpuses are small. Zumpuses are rompuses.'

output:

{"Rule1": {"conditions": ["X(is, jompus)"], "consequents": ["X(is not, shy)"]},

"Rule2": {"conditions": ["X(is, jompus)"], "consequents": ["X(is, yumpus)"]},

"Rule3": {"conditions": ["X(is, yumpus)"], "consequents": ["X(is, aggressive)"]},

[...]

"Rule17": {"conditions": ["X(is, zumpus)"], "consequents": ["X(is, rompus)"]}

}

Rules:

{{The Rule to be simplified}}

**For generating simplified representation of facts on PrONtoQA:**

You are given some known facts. Output the facts following the format of the given examples:

Examples:

Facts:

Sally is a numpus. Fae is a dumpus. Stella is shy.

Output:

{"Fact1": ["Sally(is, numpus)"], "Fact2": ["Fae(is, dumpus)"], "Fact3": ["Stella(is, shy)"]}

Facts:

{{The Facts to be simplified}}

Output:

**For generating simplified representation of question on PrONtoQA:**

You are given some known facts. Output the facts following the format of the given examples:

Examples:

Facts:

Max is sour

Output:

["Max(is, sour)", "Max(is, sour)"]

Facts:

Wren is not metallic.

Output:

["Wren(is not, metallic)", "Wren(is, metallic)"]

Facts:

Fae is not opaque.

Output:

["Fae(is not, opaque)", "Fae(is, opaque)"]

Facts:

Alex is hot.

Output:

["Alex(is, hot)", "Alex(is not, hot)"]

Facts:

Output:

**For reasoning on PrONtoQA:**

Given a context consists of some known facts and rules, think step by step is the given statement true or false.

Example:

Context:

Max is a yumpus. Each yumpus is a dumpus. Dumpuses are wumpuses. Every wumpus is an impus. Impuses are tumpuses. Tumpuses are not sour.

Statement:

Max is sour.

Output:

Max is a yumpus. Each yumpus is a dumpus. Therefore Max is a dumpus. Dumpuses are wumpuses. Therefore Max is a wumpus. Every wumpus is an impus. Therefore Max is an impus. Impuses are tumpuses. Therefore Max is a tumpus. Tumpuses are not sour. Therefore Max is not sour. Therefore the given statement 'Max is sour' is False.

Cnntext:

Stella is a yumpus. Yumpuses are zumpuses. Zumpuses are impuses. Each impus is a dumpus. Each dumpus is a vumpus. Vumpuses are bright.

Statement:

Stella is bright.

Output:

Stella is a yumpus. Yumpuses are zumpuses. Therefore Stella is a zumpus. Zumpuses are impuses. Therefore Stella is an impuses. Each impus is a dumpus. Therefore Stella is a dumpus. Each dumpus is a vumpus. Therefore Stella is a vumpus. Vumpuses are bright. Therefore Stella is bright. Therefore the given statement 'Stella is bright' is True.

{{The context used to perform reasoning}}

Statement:

{{The question to be proved}}

Output:

## B   HOW WE GENERATE THE CONCEPT MAPS

---

**Algorithm 1:** Generation of the Concept Maps

**Input:** Facts $\mathcal{F}$, Rules $\mathcal{R}$

1   $\hat{\mathcal{F}}$: $\{\hat{f}_0, \hat{f}_1, ...\hat{f}_n\}$ = Simplified($\mathcal{F}$)
2   $\hat{\mathcal{R}}$: $\{\hat{r}_0, \hat{r}_1, ...\hat{r}_n\}$ = Simplified($\mathcal{R}$)
3   where $\hat{r}_i = \{"conditions" : [cdn_0, ..., cdn_n], "consequents" : [cst_0, ..., cst_n]\}$
4   **for** $\hat{r}_i$ *in* $\hat{\mathcal{R}}$ **do**
5     **for** $\hat{r}_j$ *in* $\hat{\mathcal{R}}$ **do**
6       **if** *Fulfilled($\hat{r}_i["conditions"]$, $\hat{r}_j["consequents"]$)* **then**
7         Connect($\hat{r}_i, \hat{r}_j$)
8       **end**
9     **end**
10    **for** $\hat{f}_j$ *in* $\hat{\mathcal{F}}$ **do**
11      **if** *Fulfilled($\hat{r}_i["conditions"]$, $\hat{f}_j$)* **then**
12        Connect($\hat{r}_i, \hat{f}_j$)
13      **end**
14    **end**
15 **end**

---

## C   DETAILED ERROR ANALYSIS

The proposed method consists of several steps (i.e., Generation of Concept Maps, Generation of Mind Map, Context Reconstruction and Reasoning using LLMs). We present the detailed error analysis in Table 4. Notably, since context reconstruction is based on rules and will not cause any errors, it is not included in the detailed error analysis.

Table 4: Detailed Error Analysis on d5 split of ProofWriter

| Stages | **Generation of Concept Maps** | | Generation of Mind Map | | **LLMs' reasoning** | | |
|---|---|---|---|---|---|---|---|
| | Simplifying Rules&Facts | Connecting Rules&Facts | Simplifying Question | Generating mind map | Hallucination | Multi conditions | Multi-hop planning |
| Error Cases | 8 | 13 | 0 | 3 | 10 | 12 | 20 |
| Accuracy | 98.67% (592/600) | 97.80% (579/592) | 100% (579/579) | 99.48% (576/579) | 92.71% (534/576) | | |

The error analysis is based on the results of d5 split on the ProofWriter dataset. The test set is made up of 600 samples. According to Table 4, the main failure of COP is attributed to the instability in the reasoning of LLMs even after all the previous steps are correct. We further divide these bad cases as three types: failure caused by hallucination, failure caused by multi conditions and failure caused by multi-hop planning. Examples of the three kinds are listed below:

**Example for hallucination:**

*The concise and organized context:* Anne is round. All round people are kind. Kind people are blue.Anne is round.If someone is round and blue then they are white.

*The statement to be proved:* Anne is not white

*Ground Truth:* False.

*Proof steps produced by LLMs:* Anne is round. All round people are kind. Therefore Anne is kind. Kind people are blue. Therefore Anne is blue. If someone is round and blue then they are white. Therefore Anne is white. Therefore the given statement 'Anne is not white' is True

Despite the model generate correct proof steps, it produce a wrong answer, which might be caused by hallucination.

**Example for failure caused by multi conditions:**

*The concise and organized context:* Harry is kind.All kind, red things are green.

*The statement to be proved:* Harry is not green.

*Ground Truth:* Unknown.

*Proof steps produced by LLMs:* Harry is kind. All kind, red things are green. Therefore Harry is green. Therefore the given statement 'Harry is not green' is False.

The model happens to wrongly process rules with multi conditions like "All kind, red things are green".

**Example for failure caused by multi-hop planning:**

*The concise and organized context:* The cow chases the cat. If someone chases the cat then they visit the tiger. If someone visits the tiger then they visit the bald eagle.

*The statement to be proved:* The cow does not visit the bald eagle.

*Ground Truth:* False.

*Proof steps produced by LLMs:* The cow chases the cat. If someone chases the cat then they visit the tiger. Therefore the given statement 'The cow does not visit the bald eagle' is Unknown.

The model happens to fail in proof planning for multi-hop problems even if the context is concise and organized.

The failure caused by multi-hop planning and the failure caused by hallucination can be further mitigated by decomposing the reasoning steps into smaller tasks like least-to-most prompting and the strengthening of LLMs' fundamental ability. We leave the failure caused by multi conditions as future work.

# D    EXPERIMENTS ON FOLIO

We further conduct experiments on FOLIO, a real-world logical reasoning benchmark with various type of rules. The results are listed in Table 5.

Table 5: Comparison of label accuracy on FOLIO

| Methods | Accuracy |
|---------|----------|
| Standard | 54.60 |
| CoT | 57.84 |
| Logic-LM | 61.76 |
| **COP** | **65.27** |

To adapt to FOLIO, which is more complex and contains various language patterns and rule types, COP adopts a combination of rouge scores and semantic similarity method to generate the concept maps and mind maps. With the slight adjustment, COP outperforms CoT and LogicLM while LAM-BADA is not able to work on this dataset, demonstrating the general efficacy of COP. The concise and organized context that COP provides on FOLIO facilitates the reasoning of LLMs while CoT still suffers from redundant and out-of-order context.

# E    TOKEN USAGE AND NUMBER OF INFERENCE CALLS

We present a comparison of token numbers used per question on ProofWriter dataset with different hops in Figure 6.

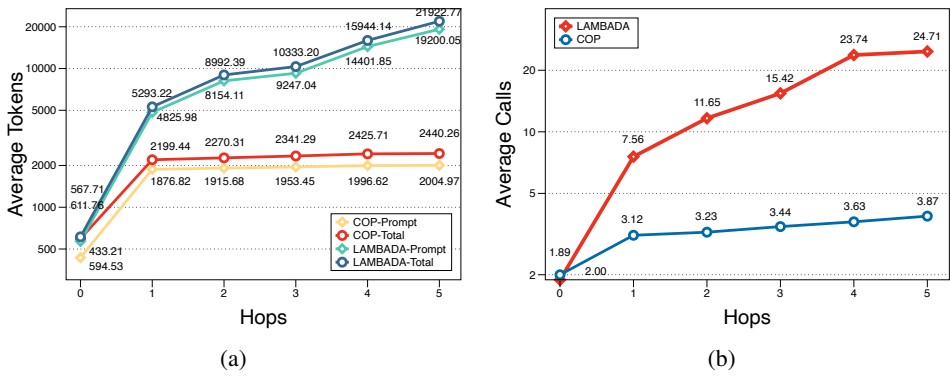

(a)                                                              (b)

Figure 6: Comparison of token usage and the number of inference calls.

The token numbers are taken from the usage statistics returned by the OpenAI API. COP-Prompt and LAMBADA-Prompt stand for the input token numbers of COP and LAMBADA while COP-Total and LAMBADA-Total stand for the overall token consumed by input and output. As is shown in the table, COP costs much fewer token numbers as well as inference calls than LAMBADA and they remain relatively stable as the number of hops increases, demonstrating our proposed COP's superiority in both effectiveness and efficiency.

# F    A BRIEF INTRODUCTION OF COP WITH A CONSISTENT RUNNING EXAMPLE

In this paper, we demonstrate that LLMs excel in deductive reasoning but struggle in proof planning. Therefore, we naturally come up with an idea to imitate of human cognition. The proposed COP first obtains a comprehensive understanding of the reasoning context by generating a concept map depicting the relevance between given rules and facts. Then, given a query that need to be proven or answered, COP identifies the most relevant information from the concept maps while eliminating redundancy, resulting in a mind map-like structure centered around the query node. After that, LLMs are prompted by the context sentences which are organized in a progressively ordered manner within one or more sequential sub-mind maps, in order to better adapt to the inference process of the model.

In this section, we briefly introduce COP based on the following example.

---

**Context:**
Rule1: All blue things are green.
Rule2: All rough, nice things are young.
Rule3: Green things are nice.
Rule4: If Erin is blue and Erin is furry then Erin is rough.
Rule5: Green, smart things are furry.
Rule6: All furry things are blue.
Fact1: Bob is furry.
Fact2: Bob is rough.
Fact3: Erin is blue.
Fact4: Erin is furry.
Fact5: Erin is green.
Fact6: Erin is nice.
Fact7: Erin is young.
**Statement to be proved:**
Bob is nice.

---

Firstly, to imitate the process of human beings organizing thoughts, a concept map is generated to present the relevance of given rules and facts. The generation process further consists of two steps.

**Simplified Representations of Rules and Facts.** To enable connecting relevant rules and facts with each other, we utilize LLMs with few-shot prompt to create a unified and simplified representation for the facts and the rules. For example, Rule1 is changed into "conditions: [X(is, blue)], consequents: [X(is, green)]" where "X" can be substituted by any entities. Fact1 is changed into "[Bob(is, furry)]".

**Connecting of Rules and Facts.** With the simplified representations of rules and facts, we connect each rule to facts as well as rules whose consequents satisfy one or more of the conditions specified in the current rule. For example, by unifying same entities, we can connect Rule1 to Fact3 since they share the same entity "blue".

Secondly, given a query (i.e., Bob is nice in the given example), we identify relevant clues from the concept maps to create a mind map with the question node at its center. The process also consists of two sub steps.

**Simplified Representations of the given question.** Similar to the simplifying process of facts and rules, we utilize LLMs with few-shot prompt to change "Bob is nice" into "[Bob(is, nice)]" and its contrary statement "[Bob(is not, nice)]".

**Generation of the mind map.** With the simplified question, we use the same way as we connect rules and facts when constructing the concept to identify the relevant rules and facts. For example, "Bob(is, nice)" can be connected by Rule3 (i.e., Green things are nice. ). Therefore, we are able to obtain a mind map by perform a D-depth searching starting from Rule3 in the concept map where D is the max reasoning depth. In this way, a number of irrelevant rules and facts can be excluded from the mind map.

The mind map might consists of several sub mind maps, each of which is a potentially possible reasoning path to determine whether the given question is True or False. Before we utilize LLMs to perform the final reasoning, we reconstruct the reasoning context in two steps:

**Sub-Mind map Pruning.** Since we know what to prove, we can remove sub mind maps which are obviously useless. Sub mind maps without a valid fact can not be used to reach a conclusion. Therefore, sub mind maps like "Rule5 -¿ Rule6 -¿ Rule1 -¿ Rule3" is removed.

**Context Reconstruction.** We reconstruct a reasoning context for each remaining sub-mind map. Given the sub mind map "Fact1 -¿ Rule6 -¿ Rule1 -¿ Rule3", the context is reconstructed as "Bob is furry. All furry things are blue. All blue things are green. Green things are nice." by traversing the sub-mind map from its leaf nodes to the root node which naturally adapts to the LLMs.

Finally, we use the reconstructed contexts to prompt the reasoning of LLMs until a true or false statement regarding the given question is made.

# G    EXAMPLES OF PROOFS GENERATED BY CoT AND COP

**Rules:**

If someone eats the bear and the bear likes the cow then the bear likes the dog. If someone is kind then they chase the mouse. If someone eats the cow then the cow is young. If someone likes the mouse then they eat the dog. If the dog likes the mouse and the mouse does not like the dog then the mouse does not like the cow. If the cow is young and the bear does not chase the cow then the cow is kind. If someone eats the cow then the cow eats the mouse. If someone eats the dog then they eat the cow.

**Facts:**

The bear does not chase the cow. The bear is big. The bear is cold. The bear is young. The bear likes the dog. The bear likes the mouse. The cow does not chase the dog. The cow likes the mouse. The dog does not chase the bear. The dog does not eat the cow. The dog is not cold. The dog does not like the bear. The dog likes the cow. The dog does not like the mouse. The mouse is kind. The mouse likes the bear.

**Question:**

The cow is not young.

**Context reconstructed by COP:**

The bear likes the mouse. The cow likes the mouse. If someone likes the mouse then they eat the dog. If someone eats the dog then they eat the cow. If someone eats the cow then the cow is young.

**LLM's proof steps produced by CoT based on the context reconstructed by COP:**

The bear likes the mouse. The cow likes the mouse. If someone likes the mouse then they eat the dog. Therefore the bear eats the dog and the cow eats the dog. If someone eats the dog then they eat the cow. Therefore the bear eats the cow and the cow eats the cow. If someone eats the cow then the cow is young. Therefore the cow is young. Therefore the given statement 'The cow is not young' is False.

**LLM's proof steps produced by CoT based on the original context:**

If the cow is not young, it means that someone has eaten the cow. According to Rule 6, if someone eats the cow, then the cow eats the mouse. However, based on the given facts, we know that the cow likes the mouse, which contradicts the statement. Therefore, the statement "The cow is not young" is False.

LLMs with CoT based on the original context can easily produce wrong answers or unfaithful proof steps, like "based on the given facts, we know that the cow likes the mouse, which contradicts the statement. " in the above example. However, with the concise and organized context reconstructed by COP, LLMs with CoT is able to produce correct answer with faithful proof steps.

