# OpenReview forum: "Concise and Organized Perception Facilitates Large Language Models for Deductive Reasoning"
_ICLR.cc/2024/Conference — Submitted to ICLR 2024_

### Official Review · Reviewer_3B4m · 2023-10-30

**Soundness:** 3 good
**Presentation:** 2 fair
**Contribution:** 1 poor
**Rating:** 3
**Confidence:** 4

**Summary:**

This paper proposes a prompting method for LLMs in synthetic logical reasoning tasks. First, the model is prompted to output a "mind-map" representation of the problem, with rules and facts represented in symbolic first-order logic terms, and then connected to each other when they share a term (like the consequent or premise of different rules match). Then, a pruning stage reduces the mind-map, eliminating irrelevant rules. Finally, the mind-maps are used in chain-of-thought reasoning about the original question. Experiments on ProofWriter and PrOntoQA, two synthetic logical reasoning datasets, show improvements for GPT-3.5-Turbo and text-davinci-003.

**Strengths:**

Experimental results are strong for the two datasets that the paper uses. The results show notable improvement on PrOntoQA-OOD with OrIntro, which is interesting.

The method is also complementary to backward chaining (e.g., LAMBADA).

CoP also takes less LLM cals than LAMBADA on average.

**Weaknesses:**

The clarity of presentation can be significantly improved. I don't fully understand the method (some questions & suggestions below).

The scope of the paper, with the current evaluations, is very narrow. The authors only use synthetic logical reasoning datasets. The problem of having lots of irrelevant context might be too specific for this particular kind of logical reasoning problem. It's unclear to me what is the takeaway for most of the ICLR audience.

Also, the authors discuss that the approach differs from LogicLM or other approaches that use an external logical solver. While I understand the difference, since here the LLM itself performs the logical reasoning and can thus be a bit more flexible with minor syntactic inconsistencies, the proposed method has some level of external symbolic processing if I understand, hand-written by the authors (e.g., partitioning the mind map by parsing the graph). Thus, it's not a "fully LLM-driven" method either.

**Questions:**

- What "misleading information" (as opposed to irrelevant) are the authors referring to in Section 3, first paragraph? In these problems, as far as I understand, everything in the question is to be taken as true (so it can at most be irrelevant).
- Method: after context reconstruction, is the input in the same format as the original input (only with rules pruned / reordered)? Or does some of the structure of the mind map still remain?
- Method: how do you unify terms that can be connected in different rules? Is it just exact match?
- How does the number of tokens used compare to LAMBADA? LLM calls are not the most important metric here, since tokens are what drive overall API costs.
- Are the results in a zero-shot or few-shot setting?
- Why did LAMBADA perform so poorly with And rules? This should be discussed.
- What other tasks other than synthetic logical reasoning problems can CoP be applied to? Any "real-world" examples? This could be datasets like FOLIO, or more broad logical reasoning tasks (e.g., LogiQA, or ReClor).
- What exactly are the "Incorrect Goal" and "Incorrect Sign" failure modes in Figure 4?
- What are the main failure modes that remain? Are they in the construction of the mind map, in parsing the rules into symbolic forms, or in reasoning even after both previous steps are correct?
- Why is LogicLM only evaluated in the 5-hop setting?

- Minor: the pseudo-code can be simplified by simply removing the "else - pass" twice

---

> ### Author Response · Authors · 2023-11-20
> **Response to Reviewer 3B4m [part 1/6]**
>
> Thank you for your careful review. We carefully discuss all your concerns and questions below.
>
> >**W1**: The clarity of presentation can be significantly improved. I don't fully understand the method.
>
> **Response for W1:**
>
> Thanks for the suggestion. We have added a consistent example in the methodology part and make some revisions to improve the clarity.
>
> ### A brief and more clear introduction of our method with a consistent running example
> In this paper, we demonstrate that LLMs excel in deductive reasoning but struggle in proof planning. Therefore, we naturally come up with an idea to imitate of human cognition. The proposed COP first obtains a comprehensive understanding of the reasoning context by generating a concept map depicting the relevance between given rules and facts. Then, given a query that need to be proven or answered, COP identifies the most relevant information from the concept maps while eliminating redundancy, resulting in a mind map-like structure centered around the query node. After that, LLMs are prompted by the context sentences which are organized in a progressively ordered manner within one or more sequential sub-mind maps, in order to better adapt to the inference process of the model.
>
> Given a context:
> ```
> Rule1: All blue things are green.
>
> Rule2: All rough, nice things are young.
>
> Rule3: Green things are nice.
>
> Rule4: If Erin is blue and Erin is furry then Erin is rough.
>
> Rule5: Green, smart things are furry.
>
> Rule6: All furry things are blue.
>
> Fact1: Bob is furry.
>
> Fact2: Bob is rough.
>
> Fact3: Erin is blue.
>
> Fact4: Erin is furry.
>
> Fact5: Erin is green.
>
> Fact6: Erin is nice.
>
> Fact7: Erin is young.
>
> ```
> The target is to prove whether it is True or False that *Bob is nice.*
>
> Firstly, to imitate the process of human beings organizing thoughts, a concept map is generated to present the relevance of given rules and facts. The generation process further consists of two steps.
> 1. **Simplified Representations of Rules and Facts.** To enable connecting relevant rules and facts with each other, we utilize LLMs with few-shot prompt to create a unified and simplified representation for the facts and the rules. For example, Rule1 is changed into “conditions: [X(is, blue)], consequents: [X(is, green)] “where “X” can be substituted by any entities. Fact1 is changed into “[Bob(is, furry)]”.
> 2. **Connecting of Rules and Facts.** With the simplified representations of rules and facts, we connect each rule to facts as well as rules whose consequents satisfy one or more of the conditions specified in the current rule. For example, by unifying same entities, we can connect Rule1 to Fact3 since they share the same entity “blue”.
>
> Secondly, given a query (i.e., Bob is nice in the given example), we identify relevant clues from the concept maps to create a mind map with the question node at its center. The process also consists of two sub steps.
>   1. **Simplified Representation of the given question.** Similar to the simplifying process of facts and rules, we utilize LLMs with few-shot prompt to change “Bob is nice” into “[Bob(is, nice)]” and its contrary statement “[Bob(is not, nice)]”.
>   2. **Generation of the mind map.** With the simplified question,  we use the same way as we connect rules and facts when constructing the concept to identify the relevant rules and facts. For example, “Bob(is, nice)” can be connected by Rule3 (i.e., Green things are nice. ). Therefore, we are able to obtain a mind map by perform a D-depth searching starting from Rule3 in the concept map where D is the max reasoning depth. In this way, a number of irrelevant rules and facts can be excluded from the mind map.
>
> The mind map might consists of several sub mind maps, each of which is a potentially possible reasoning path to determine whether the given question is True or False. Before we utilize LLMs to perform the final reasoning, we reconstruct the reasoning context in two steps:
>   1. **Sub-Mind map Pruning.** Since we know what to prove, we can remove sub mind maps which are
> obviously useless. Sub mind maps without a valid fact can not be used to reach a conclusion. Therefore, sub mind maps like “Rule5 -> Rule6 -> Rule1 -> Rule3” is removed.
>   2. **Context Reconstruction.** We reconstruct a reasoning context for each remaining sub-mind map.  Given the sub mind map “Fact1 -> Rule6 -> Rule1 -> Rule3”, the context is reconstructed as “Bob is furry. All furry things are blue. All blue things are green. Green things are nice.” by traversing the sub-mind map from its leaf nodes to the root node which naturally adapts to the LLMs.
>
> Finally, we use the reconstructed contexts to prompt the reasoning of LLMs until a true or false statement regarding the given question is made.
>
> We hope that the statements above can improve the clarity.

---

> > ### Author Response · Authors · 2023-11-20
> > **Response to Reviewer 3B4m [part 2/6]**
> >
> > >**W2**: The scope of the paper, with the current evaluations, is very narrow. The authors only use synthetic logical reasoning datasets. The problem of having lots of irrelevant context might be too specific for this particular kind of logical reasoning problem. It's unclear to me what is the takeaway for most of the ICLR audience.
> >
> > **Response for W2:**
> >
> > Thanks for the insightful suggestion. As you suggested, we have conducted experiments on FOLIO, a real-world reasoning dataset,  to address the limitation of previous evaluations performed only on the three synthetic logical reasoning datasets. With some slight modifications (details are described in Q7), COP can surpass CoT and LogicLM on FOLIO, indicating that our method can be generalized to broader and real-world logical reasoning tasks. We believe that the proposed method and the experimental findings provide valuable insights to the ICLR audience regarding broader logical reasoning.
> >
> > Our core motivations lie in more than just dealing with irrelevant context. Our observations and experimental results also demonstrate the effectiveness of an organized input on the logical reasoning performance of LLMs. Both of the two ideas have the potential to provide insightful inspiration to the community of logical reasoning.
> >
> > Moreover, our approach of carefully analyzing given statements and efficiently identifying and organizing pertinent information while eliminating redundancy, is not limited to logical reasoning alone but can be extended to more real-world scenarios involving retrieval augmented generation where a surplus of irrelevant or out-of-order information can be recalled into the context of LLMs.
> > While we believe our key insight on the proposal of easy-to-plan method has broader implications, it is much more challenging to deal with real-world scenarios. We plan to address it in future research as we stated in the conclusion and future work part of the paper.
> >
> > >**W3**: Also, the authors discuss that the approach differs from LogicLM or other approaches that use an external logical solver. While I understand the difference, since here the LLM itself performs the logical reasoning and can thus be a bit more flexible with minor syntactic inconsistencies, the proposed method has some level of external symbolic processing if I understand, hand-written by the authors (e.g., partitioning the mind map by parsing the graph). Thus, it's not a "fully LLM-driven" method either.
> >
> > **Response for W3:**
> >
> > We thank the reviewer for their careful comments on our paper. We appreciate the opportunity to address this issue.
> >
> > Firstly, our objective in developing this new method was not to create a system that relies exclusively on a large language model without any external processing. Instead, our goal was to enhance the current capabilities of LLMs in deductive reasoning by providing a more concise and organized forms of context to the model. Since LLMs struggle in proof planning, our proposed method siginificantly decreases its difficulty.
> >
> > Secondly, though our method does involve some external processing, we would like to demonstrate notable differences with LogicLM and existing neural-symbolic approaches in two key aspects.
> > 1. LogicLM utilizes LLMs to translate the context into task-specific first-order logic languages which depict the logical relationships of the original input. However, there are no logical operations involved during the generation of concept maps and mind maps in our method. It is only a process to identify and organize relevant information.
> > 2. The concept maps and mind maps are not kept once a more organized and concise form of context is obtained. We leave all the reasoning part to LLMs and the processed context is the only input to LLMs.
> >
> > Overall, by carefully adapting the given context to the inference process of LLMs in a concise and organized manner, we aim to lower the difficulty of proof planning for LLMs and unlock the deductive reasoning abilities that were previously untapped, while the external processing in our method is just an approach to obtain a concise and organized context.

---

> > > ### Author Response · Authors · 2023-11-20
> > > **Response to Reviewer 3B4m [part 3/6]**
> > >
> > > >**Q1**: What "misleading information" (as opposed to irrelevant) are the authors referring to in Section 3, first paragraph? In these problems, as far as I understand, everything in the question is to be taken as true (so it can at most be irrelevant).
> > >
> > > **Response for Q1:**
> > >
> > > Sorry for the confusions about misleading information and irrelevant information, we use an example taken from the ProofWriter dataset to clarify it as below.
> > >
> > > Given rules:
> > > Nice, smart people are cold.
> > > If someone is cold then they are smart.
> > > Rough people are cold.
> > > All nice, rough people are smart.
> > > Cold people are nice.
> > > Young people are smart.
> > > Nice people are furry.
> > > All smart people are rough.
> > > Cold, round people are furry.
> > >
> > > Erin is smart. Determine whether it is True, False or unknown that Erin is furry.
> > >
> > > The golden proof steps are “Erin is smart. All smart people are rough. Therefore, Erin is rough. Rough people are cold. Therefore, Erin is cold. Cold people are nice. Therefore, Erin is nice. Nice people are furry. Therefore, it is True that Erin is furry”.
> > >
> > > However, the rule that “Cold, round people are furry.” can mislead LLMs to predict the result as unknown since there is no information about whether Erin is round or not. Such rules significantly increase the difficulty of proof planning, resulting in failure of reasoning. Therefore, we define rules that can be relevant in some steps of reasoning yet leading to wrong answers and not included in the gold proof steps as “misleading”.
> > >
> > > >**Q2**: Method: after context reconstruction, is the input in the same format as the original input (only with rules pruned / reordered)? Or does some of the structure of the mind map still remain?
> > >
> > > **Response for Q2:**
> > >
> > > The structure of mind map will not remain after context reconstruction.
> > > In most cases, the input is only reconstructed with rules pruned and reordered. In some hard cases, however, some facts and rules repeatedly appear in the mind map as they are used multi times in the multi-hop reasoning steps. While they only appear once in the original input, we duplicate them when reconstructing the context following the structure of the mind map. Below is an example:
> > >
> > > Original context:
> > > Smart, cold things are white. If Dave is green then Dave is not blue. If Erin is green then Erin is smart. If something is white and furry then it is smart. If Fiona is cold and Fiona is quiet then Fiona is smart. If something is furry then it is green. All green, white things are furry. If something is white and quiet then it is furry. Green, quiet things are blue.
> > >
> > > Dave is not cold. Erin is cold. Erin is furry. **Fiona is cold.** **Fiona is quiet.** Harry is smart. Harry is white.
> > >
> > > The target is to determine whether it is True or False that Fiona is not green.
> > >
> > > Reconstructed context:
> > > **Fiona is cold.** **Fiona is quiet.** If Fiona is cold and Fiona is quiet then Fiona is smart. **Fiona is cold.** Smart, cold things are white. **Fiona is quiet.** If something is white and quiet then it is furry. If something is furry then it is green.
> > >
> > > The facts that Fiona is cold and Fiona is quiet are duplicated to make sure they are repeated whenever they are used by a rule, thus to improve the reasoning performance of LLMs.
> > >
> > > >**Q3**: Method: how do you unify terms that can be connected in different rules? Is it just exact match?
> > >
> > > **Response for Q3:**
> > >
> > > That’s an insightful question. Besides exact match which is the most straightforward method, we have adopt other methods such as using rouge scores, using semantic similarity based on vector representations or even using LLMs to select relevant terms. The results of experiments on the three synthetic datasets are based on exact match because exact match can achieve comparable performance with other methods on theses datasets. For the FOLIO dataset you suggested, we adopt a combination of rouge scores and semantic similarity method to unify terms. The detailed experimental results of FOLIO are stated in **Q7**.

---

> > > > ### Author Response · Authors · 2023-11-20
> > > > **Response to Reviewer 3B4m [part 4/6]**
> > > >
> > > > >**Q4**: How does the number of tokens used compare to LAMBADA? LLM calls are not the most important metric here, since tokens are what drive overall API costs.
> > > >
> > > > **Response for Q4:**
> > > >
> > > > Thanks for the helpful comment. We add a comparison of token numbers used per question on ProofWriter dataset with different hops as below.
> > > >
> > > > | Hops  | 0  | 1  | 2  | 3 | 4  | 5  |
> > > > |---|---|---|---|---|---|---|
> > > > | LAMBADA-Prompt   | 567.71  | 4825.98  | 8154.11  | 9247.04  | 14401.85  | 19200.05  |
> > > > |  LAMBADA-Total  |  611.76 | 5293.22  | 8992.39  | 10333.20  | 15944.14  | 21922.77  |
> > > > |  COP-Prompt  | 433.21  | 1876.82  | 1915.68  | 1953.45  | 1996.62  | 2004.97  |
> > > > |  COP-Total  | 594.53  | 2199.44  | 2270.31  |  2341.29 | 2425.71  | 2440.26  |
> > > >
> > > > The token numbers are taken from the usage statistics returned by the OpenAI API. COP-Prompt and LAMBADA-Prompt stand for the input token numbers of COP and LAMBADA while COP-Total and LAMBADA-Total stand for the overall token consumed by input and output.
> > > > As is shown in the table, COP costs much fewer token numbers than LAMBADA and the number of token numbers remain relatively stable as the number of hops increases, demonstrating our proposed COP’s superiority in both effectiveness and efficiency.
> > > >
> > > > >**Q5**: Are the results in a zero-shot or few-shot setting?
> > > >
> > > > **Response for Q5:**
> > > >
> > > > For a fair comparison, we used few-shot setting following LAMBADA and Selection inference. And the few-shot examples we used are listed in the prompt which can be find in the Appendix.
> > > >
> > > > >**Q6**: Why did LAMBADA perform so poorly with And rules? This should be discussed.
> > > >
> > > > **Response for Q6:**
> > > >
> > > > Thanks for the suggestion.
> > > > "And" rules have two types, including "AndIntro" and "AndElim". The statement that should be proved in "AndElim" part is "A is b." (For example, Max is a lorpus.). The statement that should be proved in "AndIntro" part is "A is b and c." (For example, Sally is luminous and a lorpus.). Besides, there are rules of "AndIntro" type in "AndElim" part.  LAMBADA fails in Goal Decomposition step to handle the rules of "AndIntro" type, resulting in poor performance with "And" rules.
> > > >
> > > > >**Q7**: What other tasks other than synthetic logical reasoning problems can CoP be applied to? Any "real-world" examples? This could be datasets like FOLIO, or more broad logical reasoning tasks (e.g., LogiQA, or ReClor).
> > > >
> > > > **Response for Q7:**
> > > >
> > > > As you suggested, we further conduct experiments on FOLIO, a real-world logical reasoning benchmark with various type of rules. Datasets like LogiQA and ReClor, though containing logical reasoning, involve machine reading comprehension which is not included in our current scope, therefore, we leave experiments on these two datasets for future work.
> > > >
> > > > To adapt to FOLIO, which is more complex and contains various language patterns and rule types, COP adopts a combination of rouge scores and semantic similarity method to generate the concept maps and mind maps. With the slight adjustment, COP outperforms CoT and LogicLM while LAMBADA is not able to work on this dataset, demonstrating the general efficacy of COP. The concise and organized context that COP provides on FOLIO facilitates the reasoning of LLMs while CoT still suffers from redundant and out-of-order context.
> > > >
> > > > | Methods  | Accuracy  |
> > > > |---|---|
> > > > |  Standard | 54.60  |
> > > > | CoT  | 57.84  |
> > > > | Logic-LM  | 61.76  |
> > > > | **COP**  | **65.27**  |
> > > >
> > > > >**Q8**: What exactly are the "Incorrect Goal" and "Incorrect Sign" failure modes in Figure 4?
> > > >
> > > > **Response for Q8:**
> > > >
> > > > There are four steps in LAMBADA, which are Fact Check step, Rule Selection step, Goal Decomposition step and Sign Agreement step. "Incorrect Goal" means that LAMBADA fails in Goal Decomposition steps. "Incorrect Sign" means that LAMBADA fails in Sign Agreement steps.

---

> > > > > ### Author Response · Authors · 2023-11-20
> > > > > **Response to Reviewer 3B4m [part 5/6]**
> > > > >
> > > > > >**Q9**: What are the main failure modes that remain? Are they in the construction of the mind map, in parsing the rules into symbolic forms, or in reasoning even after both previous steps are correct?
> > > > >
> > > > > **Response for Q9:**
> > > > >
> > > > > To address your concerns, we add the detailed error analysis as suggested. The proposed method consists of several steps (i.e., Generation of Concept Maps, Generation of Mind Map, Context Reconstruction and Reasoning using LLMs).  Notably, since context reconstruction is based on rules  and will not cause any errors, it is not included in the detailed error analysis.
> > > > >
> > > > > |   | **Generation of Concept Maps** | **Generation of Concept Maps** |  **Generation of Mind Map** | **Generation of Mind Map**  | **LLMs’ reasoning** |  **LLMs’ reasoning** | **LLMs’ reasoning**  |
> > > > > |---|:-:|:-:|:-:|:-:|:-:|:-:|:-:|
> > > > > |   |  Simplified Representation of Rules and Facts |  Connection of Rules and Facts |  Simplified Representation of Question  | Generation of mind map  | hallucination  |  Multi conditions |  Multi-hop planning |
> > > > > |  Number of Error Cases |  8  |  13 | 0  | 3  | 10  | 12  | 20  |
> > > > > |  Accuracy |  98.67% (592/600)  | 97.80% (579/592)  |  100% (579/576 /579) | 99.48% (576/579)  | 92.71% (534/576) | 92.71% (534/576) | 92.71% (534/576) |
> > > > >
> > > > > The main failure of COP lies in reasoning even after all the previous steps are correct. We further divide these bad cases as three types: failure caused by hallucination, failure caused by multi conditions and failure caused by multi-hop planning. Examples of the three kinds are listed below:
> > > > >
> > > > > ### Example for hallucination:
> > > > >
> > > > > **The concise and organized context:** Anne is round. All round people are kind. Kind people are blue.Anne is round.If someone is round and blue then they are white.
> > > > >
> > > > > **The statement to determine whether it is True or False**: Anne is not white
> > > > >
> > > > > **Ground Truth:** False.
> > > > >
> > > > > **Proof steps produced by LLMs:** Anne is round. All round people are kind. Therefore Anne is kind. Kind people are blue. Therefore Anne is blue. If someone is round and blue then they are white. Therefore Anne is white. Therefore the given statement 'Anne is not white' is True
> > > > >
> > > > > Despite the model generate correct proof steps, it produce a wrong answer, which might be caused by hallucination.
> > > > >
> > > > > ### Example for failure caused by multi conditions:
> > > > >
> > > > > **The concise and organized context:** Harry is kind.All kind, red things are green.
> > > > >
> > > > > **The statement to determine whether it is True or False**: Harry is not green.
> > > > >
> > > > > **Ground Truth:** Unknown.
> > > > >
> > > > > **Proof steps produced by LLMs:** Harry is kind. All kind, red things are green. Therefore Harry is green. Therefore the given statement 'Harry is not green' is False.
> > > > >
> > > > > The model happens to wrongly process rules with multi conditions like “All kind, red things are green”.
> > > > >
> > > > > ### Example for failure caused by multi-hop planning:
> > > > >
> > > > > **The concise and organized context:** The cow chases the cat. If someone chases the cat then they visit the tiger. If someone visits the tiger then they visit the bald eagle.
> > > > >
> > > > > **The statement to determine whether it is True or False**: The cow does not visit the bald eagle.
> > > > >
> > > > > **Ground Truth:** False.
> > > > >
> > > > > **Proof steps produced by LLMs:** The cow chases the cat. If someone chases the cat then they visit the tiger. Therefore the given statement 'The cow does not visit the bald eagle' is Unknown.
> > > > >
> > > > > The model happens to failure of proof planning for multi-hop problems even if the context is concise and organized.
> > > > >
> > > > > The failure caused by multi-hop planning and the failure caused by hallucination can be further mitigated by decomposing the reasoning steps into smaller tasks like least-to-most prompting and the strengthening of LLMs’ fundamental ability. We leave the failure caused by multi conditions as future work.

---

> > > > > > ### Author Response · Authors · 2023-11-20
> > > > > > **Response to Reviewer 3B4m [part 6/6]**
> > > > > >
> > > > > > >**Q10**: Why is LogicLM only evaluated in the 5-hop setting?
> > > > > >
> > > > > > **Response for Q10:**
> > > > > >
> > > > > > The results of LogicLM are taken from its original paper. As the authors of LogicLM stated, they only evaluated LogicLM in the 5-hop settings because it is the hardest setting. In fact, we try to reproduce the results of LogicLM and conduct experiments on other settings based on the official code released by its authors. However, the reproduced results are lower than that reported in its paper and show significant gap with the proposed method in other settings. Therefore, to avoid any confusions, we chose to only contain the reported results of LogicLM in our paper.
> > > > > > Below we give the reproduced results of LogicLM on all of our experimental settings. The code used to conduct these experiments as well as the implementation of our proposed method will be made publicly available.
> > > > > >
> > > > > > |   | ProofWriter  | ProofWriter  |  ProofWriter |  ProofWriter | ProofWriter  | PrOntoQA  |  PrOntoQA |  PrOntoQA |
> > > > > > |---|---|---|---|---|---|---|---|---|
> > > > > > |   |  d5 | d3  | d2  | d1  | d0  | 5-hop  | 3-hop  | 1-hop  |
> > > > > > | CoT | 53.50  | 61.17  | 61.33  | 62.33  | 62.83  | 69.80  | 74.20  | 86.20  |
> > > > > > |  SI |  46.00 | 51.00  | 56.00  | 61.00  | 97.00  | 45.00  | 52.00  | 97.00  |
> > > > > > |  LogicLM | 65.11  | 66.95  | 70.24  | 68.25  | 74.10  | 58.48  | 56.62  | 61.28  |
> > > > > > | LAMBADA  | 72.00  | 82.00  | 87.00  | 90.00  | 98.00  | 96.00  |  99.00 | 98.00  |
> > > > > > | **COP**  | **88.67**  | **90.67**  | **91.43**  | **92.50**  | **98.50**  | **99.20**  | **99.60**  | **100.00**  |
> > > > > >
> > > > > > >**Q11**: Minor: the pseudo-code can be simplified by simply removing the "else - pass" twice
> > > > > >
> > > > > > **Response for Q11:**
> > > > > >
> > > > > > Thanks for the kind suggestion. We have modified this part as suggested.

---

> > > > > > ### Author Response · Authors · 2023-11-20
> > > > > > **Response to Reviewer 3B4m [part 6/6]**
> > > > > >
> > > > > > >**Q10**: Why is LogicLM only evaluated in the 5-hop setting?
> > > > > >
> > > > > > **Response for Q10:**
> > > > > >
> > > > > > The results of LogicLM are taken from its original paper. As the authors of LogicLM stated, they only evaluated LogicLM in the 5-hop settings because it is the hardest setting. In fact, we try to reproduce the results of LogicLM and conduct experiments on other settings based on the official code released by its authors. However, the reproduced results are lower than that reported in its paper and show significant gap with the proposed method in other settings. Therefore, to avoid any confusions, we chose to only contain the reported results of LogicLM in our paper.
> > > > > > Below we give the reproduced results of LogicLM on all of our experimental settings. The code used to conduct these experiments as well as the implementation of our proposed method will be made publicly available.
> > > > > >
> > > > > > |   | ProofWriter  | ProofWriter  |  ProofWriter |  ProofWriter | ProofWriter  | PrOntoQA  |  PrOntoQA |  PrOntoQA |
> > > > > > |---|---|---|---|---|---|---|---|---|
> > > > > > |   |  d5 | d3  | d2  | d1  | d0  | 5-hop  | 3-hop  | 1-hop  |
> > > > > > | CoT | 53.50  | 61.17  | 61.33  | 62.33  | 62.83  | 69.80  | 74.20  | 86.20  |
> > > > > > |  SI |  46.00 | 51.00  | 56.00  | 61.00  | 97.00  | 45.00  | 52.00  | 97.00  |
> > > > > > |  LogicLM | 65.11  | 66.95  | 70.24  | 68.25  | 74.10  | 58.48  | 56.62  | 61.28  |
> > > > > > | LAMBADA  | 72.00  | 82.00  | 87.00  | 90.00  | 98.00  | 96.00  |  99.00 | 98.00  |
> > > > > > | **COP**  | **88.67**  | **90.67**  | **91.43**  | **92.50**  | **98.50**  | **99.20**  | **99.60**  | **100.00**  |
> > > > > >
> > > > > > >**Q11**: Minor: the pseudo-code can be simplified by simply removing the "else - pass" twice
> > > > > >
> > > > > > **Response for Q11:**
> > > > > >
> > > > > > Thanks for the kind suggestion. We have modified this part as suggested.

---

### Official Review · Reviewer_Fc11 · 2023-10-30

**Soundness:** 4 excellent
**Presentation:** 4 excellent
**Contribution:** 3 good
**Rating:** 8
**Confidence:** 3

**Summary:**

This work studies complex deductive reasoning problems for large language models. They aim to reduce the difficulty of LLMs proof planning and propose a reasoning approach named Concise and Organized Perception (COP). The approach has three stages. It first creates concept maps to highlight the hierarchical relationships among the provided rules and facts, then identifies the relevant contexts and generates a mind map-like structure based on the provided question, and finally prunes the mind map and reconstructs the context for prompting the reasoning of LLMs until a true or false statement regarding the given question is made. They conduct experiments on three synthetic logical reasoning datasets and demonstrate that the approach is effective and efficient.

**Strengths:**

The proposed COP outperforms the compared SoTA methods in different deductive rules. It greatly relieves the problem of CoT predicting a correct label with incorrect reasoning chains according to the manual check. COP can also adapt to different LLMs and be more efficient in inference calls.
The paper is well-organized and clearly written. It includes comprehensive experiments, and the results support their claim. The proposed method and the experimental findings provide valuable thoughts to the LLMs deductive reasoning problem.

**Weaknesses:**

Please see the questions listed below.

**Questions:**

Q1: What are the proofs generated by CoT are like? How does CoT concretely improve the proof planning of LLMs? It would be better if the authors could show some of the cases.

Q2: The experiments are conducted on three synthetical logical reasoning datasets. I wonder if this approach can be adapted to real-world data.

---

> ### Author Response · Authors · 2023-11-20
> **Response to Reviewer Fc11**
>
> Thanks for your insightful feedback. We are encouraged that you find our paper valuable and each of your questions are discussed below.
>
> >**Q1**: What are the proofs generated by CoT are like? How does CoT concretely improve the proof planning of LLMs? It would be better if the authors could show some of the cases.
>
> **Response for Q1:**
>
> Thanks for the kind suggestion. We suppose that you are asking about how does COP rather than CoT improve the proof planning of LLMs. In some cases, LLMs with CoT based on the original context fail to find the optimal facts and rules to determine whether the given statement is True or False. In some other cases, LLMs with CoT can obtain the correct answer with incorrect reasoning chains. Due to the word limit, we show one case below.
>
> Rules:
>
> If someone eats the bear and the bear likes the cow then the bear likes the dog. If someone is kind then they chase the mouse. If someone eats the cow then the cow is young. If someone likes the mouse then they eat the dog. If the dog likes the mouse and the mouse does not like the dog then the mouse does not like the cow. If the cow is young and the bear does not chase the cow then the cow is kind. If someone eats the cow then the cow eats the mouse. If someone eats the dog then they eat the cow.
>
> Facts:
>
> The bear does not chase the cow. The bear is big. The bear is cold. The bear is young. The bear likes the dog. The bear likes the mouse. The cow does not chase the dog. The cow likes the mouse. The dog does not chase the bear. The dog does not eat the cow. The dog is not cold. The dog does not like the bear. The dog likes the cow. The dog does not like the mouse. The mouse is kind. The mouse likes the bear.
>
> Question:
>
> The cow is not young.
>
> **Context reconstructed by COP:**
> The bear likes the mouse. The cow likes the mouse. If someone likes the mouse then they eat the dog. If someone eats the dog then they eat the cow. If someone eats the cow then the cow is young.
>
> **LLM’s proof steps produced by CoT based on the context reconstructed by COP:**
> The bear likes the mouse. The cow likes the mouse. If someone likes the mouse then they eat the dog. Therefore the bear eats the dog and the cow eats the dog. If someone eats the dog then they eat the cow. Therefore the bear eats the cow and the cow eats the cow. If someone eats the cow then the cow is young. Therefore the cow is young. Therefore the given statement 'The cow is not young' is False.
>
> **LLM’s proof steps produced by CoT based on the original context:**
> If the cow is not young, it means that someone has eaten the cow. According to Rule 6, if someone eats the cow, then the cow eats the mouse. However, based on the given facts, we know that the cow likes the mouse, which contradicts the statement. Therefore, the statement "The cow is not young" is False.
>
>
> LLMs with CoT based on the original context can easily produce wrong answers or unfaithful proof steps, like “based on the given facts, we know that the cow likes the mouse, which contradicts the statement. ” in the above example. However, with the concise and organized context reconstructed by COP, LLMs with CoT is able to produce correct answer with faithful proof steps.
>
> >**Q2**: The experiments are conducted on three synthetical logical reasoning datasets. I wonder if this approach can be adapted to real-world data.
>
> **Response for Q2:**
>
> Yes, COP can be adapted to real-world data.
> To address this issue, we further conduct experiments on FOLIO, a real-world logical reasoning benchmark with various type of rules. To adapt to FOLIO, which is more complex and contains various language patterns and rule types, COP adopts a combination of rouge scores and semantic similarity method to generate the concept maps and mind maps. With the slight adjustment, COP outperforms CoT and LogicLM while LAMBADA is not able to work on this dataset, demonstrating the general efficacy of COP. The concise and organized context that COP provides on FOLIO facilitates the reasoning of LLMs while CoT still suffers from redundant and out-of-order context.
>
> | Methods  | Accuracy  |
> |---|---|
> |  Standard | 54.60  |
> | CoT  | 57.84  |
> | Logic-LM  | 61.76  |
> | **COP**  | **65.27**  |

---

### Official Review · Reviewer_pmsh · 2023-11-01

**Soundness:** 3 good
**Presentation:** 3 good
**Contribution:** 3 good
**Rating:** 6
**Confidence:** 2

**Summary:**

This work proposes a novel technique on distilling and organizing relevant fact for reasoning. This technique can not only reduce the cost of querying the large language models but also lead to better reasoning performance comparing to the existing state of the art baselines.

**Strengths:**

Originality: 3.5/5

Although there are a few works on decomposing a question into smaller pieces where the language models are better at solving, this work focuses on distilling related knowledge to optimize the number of queries to large language models.

Quality: 4/5

The work has been evaluated on a handful of datasets and greatly improved the existing baselines in both accuracy and query number to language models.

Clarity: 2.5/5

The methodology part of the concept, mind graph generation, and mind graph pruning is a bit hard to read without a consistent running example and the prompt.

Significance: 3/5

As the LLM query is priced based on token numbers, reducing the cost and improving the performance is out of people's interest.

**Weaknesses:**

See Strength.

**Questions:**

1. What is the failure case analysis on the concept graph generation, mind graph generation, and mind graph pruning?

2. Related work: Scallop[1] is a similar approach as logic-LM, but with probabilistic reasoning engine.

[1] Hanlin Zhang, Jiani Huang, Ziyang Li, Mayur Naik, and Eric Xing. 2023. Improved Logical Reasoning of Language Models via Differentiable Symbolic Programming.

---

> ### Author Response · Authors · 2023-11-20
> **Response to Reviewer pmsh [part 1/3]**
>
> Thanks for the thoughtful and constructive suggestions. We discuss each of them below and hope that the changes made can improve the clarity and significance of this work.
>
> >**W1**: The methodology part of the concept, mind graph generation, and mind graph pruning is a bit hard to read without a consistent running example and the prompt.
>
> **Response for W1:**
>
> Thanks for the kind suggestion. We have added a consistent running example as suggested and hope it can improve the clarity.
>
> ### A brief and more clear introdution of our method with a consistent running example
> In this paper, we demonstrate that LLMs excel in deductive reasoning but struggle in proof planning. Therefore, we natually come up with an idea to imitate of human cognition. The proposed COP first obtains a comprehensive understanding of the reasoning context by generating a concept map depicting the relevance between given rules and facts. Then, given a query that need to be proven or answered, COP identifies the most relevant information from the concept maps while eliminating redundancy, resulting in a mind map-like structure centered around the query node. After that, LLMs are prompted by the context sentences which are organized in a progressively ordered manner within one or more sequential sub-mind maps, in order to better adapt to the inference process of the model.
>
> Given a context:
> ```
> Rule1: All blue things are green.
>
> Rule2: All rough, nice things are young.
>
> Rule3: Green things are nice.
>
> Rule4: If Erin is blue and Erin is furry then Erin is rough.
>
> Rule5: Green, smart things are furry.
>
> Rule6: All furry things are blue.
>
> Fact1: Bob is furry.
>
> Fact2: Bob is rough.
>
> Fact3: Erin is blue.
>
> Fact4: Erin is furry.
>
> Fact5: Erin is green.
>
> Fact6: Erin is nice.
>
> Fact7: Erin is young.
>
> ```
> The target is to prove whether it is True or False that *Bob is nice.*
>
> Firstly, to imitate the process of human beings organizing thoughts, a concept map is generated to present the relevance of given rules and facts. The generation process further consists of two steps.
> 1. **Simplified Representations of Rules and Facts.** To enable connecting relevant rules and facts with each other, we utilize LLMs with few-shot prompt to create a unified and simplified representation for the facts and the rules. For example, Rule1 is changed into “conditions: [X(is, blue)], consequents: [X(is, green)] “where “X” can be substituted by any entities. Fact1 is changed into “[Bob(is, furry)]”.
> 2. **Connecting of Rules and Facts.** With the simplified representations of rules and facts, we connect each rule to facts as well as rules whose consequents satisfy one or more of the conditions specified in the current rule. For example, by unifying same entities, we can connect Rule1 to Fact3 since they share the same entity “blue”.
>
> Secondly, given a query (i.e., Bob is nice in the given example), we identify relevant clues from the concept maps to create a mind map with the question node at its center. The process also consists of two sub steps.
>   1. **Simplified Representation of the given question.** Similar to the simplifying process of facts and rules, we utilize LLMs with few-shot prompt to change “Bob is nice” into “[Bob(is, nice)]” and its contrary statement “[Bob(is not, nice)]”.
>   2. **Generation of the mind map.** With the simplified question,  we use the same way as we connect rules and facts when constructing the concept to identify the relevant rules and facts. For example, “Bob(is, nice)” can be connected by Rule3 (i.e., Green things are nice. ). Therefore, we are able to obtain a mind map by perform a D-depth searching starting from Rule3 in the concept map where D is the max reasoning depth. In this way, a number of irrelevant rules and facts can be excluded from the mind map.
>
> The mind map might consists of several sub mind maps, each of which is a potentially possible reasoning path to determine whether the given question is True or False. Before we utlize LLMs to perform the final reasoning, we reconstruct the reasoning context in two steps:
>   1. **Sub-Mind map Pruning.** Since we know what to prove, we can remove sub mind maps which are
> obviously useless. Sub mind maps without a valid fact can not be used to reach a conclusion. Therefore, sub mind maps like “Rule5 -> Rule6 -> Rule1 -> Rule3” is removed.
>   2. **Context Reconstruction.** We reconstruct a reasoning context for each remaining sub-mind map.  Given the sub mind map “Fact1 -> Rule6 -> Rule1 -> Rule3”, the context is reconstructed as “Bob is furry. All furry things are blue. All blue things are green. Green things are nice.” by traversing the sub-mind map from its leaf nodes to the root node which naturally adapts to the LLMs.
>
> Finally, we use the reconstructed contexts to prompt the reasoning of LLMs until a true or false statement regarding the given question is made.
>
> We hope that the above statements can improve the clarity.

---

> > ### Author Response · Authors · 2023-11-20
> > **Response to Reviewer pmsh [part 2/3]**
> >
> > >**W2**: As the LLM query is priced based on token numbers, reducing the cost and improving the performance is out of people's interest.
> >
> > **Response for W2:**
> >
> > That is a very insightful comment!  We add a comparison of token numbers used per question on ProofWriter dataset with different hops as suggested.
> >
> > | Hops  | 0  | 1  | 2  | 3 | 4  | 5  |
> > |---|---|---|---|---|---|---|
> > | LAMBADA-Prompt   | 567.71  | 4825.98  | 8154.11  | 9247.04  | 14401.85  | 19200.05  |
> > |  LAMBADA-Total  |  611.76 | 5293.22  | 8992.39  | 10333.20  | 15944.14  | 21922.77  |
> > |  COP-Prompt  | 433.21  | 1876.82  | 1915.68  | 1953.45  | 1996.62  | 2004.97  |
> > |  COP-Total  | 594.53  | 2199.44  | 2270.31  |  2341.29 | 2425.71  | 2440.26  |
> >
> > The token numbers are taken from the usage statistics returned by the OpenAI API. COP-Prompt and LAMBADA-Prompt stand for the input token numbers of COP and LAMBADA while COP-Total and LAMBADA-Total stand for the overall token consumed by input and output.
> > As is shown in the table, COP costs much fewer token numbers than LAMBADA and the number of token numbers remain relatively stable as the number of hops increases, demonstrating our proposed COP’s superiority in both effectiveness and efficiency.
> >
> > >**Q1**: What is the failure case analysis on the concept graph generation, mind graph generation, and mind graph pruning?
> >
> > **Response for Q1:**
> >
> > We add the detailed error analysis in the experiment section as suggested.
> >
> > The proposed method consists of several steps (i.e., Generation of Concept Maps, Generation of Mind Map, Context Reconstruction and Reasoning using LLMs). Notably, since context reconstruction, including mind graph pruning, is based on rules and will not cause any errors, it is not included in the detailed error analysis.
> >
> > |   | **Generation of Concept Maps** | **Generation of Concept Maps** |  **Generation of Mind Map** | **Generation of Mind Map**  | **LLMs’ reasoning** |  **LLMs’ reasoning** | **LLMs’ reasoning**  |
> > |---|:-:|:-:|:-:|:-:|:-:|:-:|:-:|
> > |   |  Simplified Representation of Rules and Facts |  Connection of Rules and Facts |  Simplified Representation of Question  | Generation of mind map  | hallucination  |  Multi conditions |  Multi-hop planning |
> > |  Number of Error Cases |  8  |  13 | 0  | 3  | 10  | 12  | 20  |
> > |  Accuracy |  98.67% (592/600)  | 97.80% (579/592)  |  100% (579/576 /579) | 99.48% (576/579)  | 92.71% (534/576) | 92.71% (534/576) | 92.71% (534/576) |
> >
> > The error analysis is based on the results of d5 split on the ProofWriter dataset. The test set is made up of 600 samples. According to the table, the main failure of COP is attributed to the instability in the reasoning of LLMs even after all the previous steps are correct. We further divide these bad cases as three types: failure caused by hallucination, multi conditions and multi-hop planning. Examples of the three kinds are listed below:
> >
> > ### Example for hallucination:
> >
> > **The concise and organized context:** Anne is round. All round people are kind. Kind people are blue.Anne is round.If someone is round and blue then they are white.
> >
> > **The statement to be proved**: Anne is not white
> >
> > **Ground Truth:** False.
> >
> > **Proof steps produced by LLMs:** Anne is round. All round people are kind. Therefore Anne is kind. Kind people are blue. Therefore Anne is blue. If someone is round and blue then they are white. Therefore Anne is white. Therefore the given statement 'Anne is not white' is True
> >
> > Despite the model generate correct proof steps, it produce a wrong answer, which might be caused by hallucination.
> >
> > ### Example for failure caused by multi conditions:
> >
> > **The concise and organized context:** Harry is kind.All kind, red things are green.
> >
> > **The statement to be proved**: Harry is not green.
> >
> > **Ground Truth:** Unknown.
> >
> > **Proof steps produced by LLMs:** Harry is kind. All kind, red things are green. Therefore Harry is green. Therefore the given statement 'Harry is not green' is False.
> >
> > The model happens to wrongly process rules with multi conditions like “All kind, red things are green”.
> >
> > ### Example for failure caused by multi-hop planning:
> >
> > **The concise and organized context:** The cow chases the cat. If someone chases the cat then they visit the tiger. If someone visits the tiger then they visit the bald eagle.
> >
> > **The statement to be proved**: The cow does not visit the bald eagle.
> >
> > **Ground Truth:** False.
> >
> > **Proof steps produced by LLMs:** The cow chases the cat. If someone chases the cat then they visit the tiger. Therefore the given statement 'The cow does not visit the bald eagle' is Unknown.
> >
> > The model happens to failure of proof planning for multi-hop problems even if the context is concise and organized.
> >
> > The failure caused by multi-hop planning and the failure caused by hallucination can be further mitigated by decomposing the reasoning steps into smaller tasks like least-to-most prompting and the strengthening of LLMs’ fundamental ability. We leave the failure caused by multi conditions as future work.

---

> > > ### Author Response · Authors · 2023-11-20
> > > **Response to Reviewer pmsh [part 3/3]**
> > >
> > > >**Q2**: Related work: Scallop[1] is a similar approach as logic-LM, but with probabilistic reasoning engine.
> > >
> > > **Response for Q2:**
> > >
> > > Thanks for the kind reminder. We have already carefully analysed the paper in our related work.

---

> ### Comment · Reviewer_pmsh · 2023-11-21
> **Updates on my review**
>
> After reading the author rebuttal, I am convinced by the new data and new story. I have raised my review score from 5 to 6.

---

> > ### Author Response · Authors · 2023-11-22
> >
> > Thank you for adjusting the score and for your valuable comments to enhance the quality of our work.

---

### Official Review · Reviewer_fpHM · 2023-11-05

**Soundness:** 2 fair
**Presentation:** 2 fair
**Contribution:** 2 fair
**Rating:** 5
**Confidence:** 3

**Summary:**

The paper presents a novel approach titled Concise and Organized Perception (COP) that aims to enhance the deductive reasoning capabilities of Large Language Models (LLMs). They show that while LLMs like GPT-3 have shown promise in complex reasoning tasks, they often struggle with systematic reasoning and tend to produce errors due to misaligned information flow and lack of hierarchical understanding. They observe that the structured approach of COP can reduce the difficulty of multi-hop reasoning tasks for LLMs, potentially leading to more accurate and efficient processing of complex deductive reasoning questions.

**Strengths:**

1. The COP approach takes advantage of both "Concise" and "Organized" strategies to improve LLMs' deductive reasoning capabilities, a technique that hasn't been explored extensively in previous works. This approach also involves generating mind maps and reconstructing context, which is a unique integration of visualized reasoning into LLMs.
2. The COP's extension to pre-existing models like ProofWriter and ProtoQA demonstrates a level of creativity in enhancing the capabilities of LLMs beyond simple iterative refinements, potentially indicating a new direction for future LLM-based deductive reasoning research.
3. The results show a significant improvement over established baselines, with the COP method outperforming traditional methods and even newer approaches like LAMBADA+COP, especially in multi-hop reasoning problems.

**Weaknesses:**

1. The paper seems to heavily build upon pre-existing methods such as "graph-based reasoning systems" and "structured knowledge integration," which have been extensively explored in the literature (e.g.,logical NNs and other neuro-symbolic approaches)
2. The justification for why the combination of concise and organized strategies is more effective remains unclear under a rigorous theoretical framework. The examples provided (Sec 3 and Fig 3) do not offer evidence that this combination offers a qualitatively different approach to reasoning in LLMs as compared to existing methods.
3. The experimental results reported in Section 4.2 might not be robust enough. The benchmarks ProofWriter and ProtoQA, is not sufficiently demonstrating the general efficacy of COP across various datasets and reasoning tasks. Besides, the lack of a detailed error analysis or ablation study also keep the improvement of the whole system a mystery.

**Questions:**

Illustrated in the weaknesses.

---

> ### Author Response · Authors · 2023-11-20
> **Response to Reviewer fpHM [part 1/2]**
>
> Thanks for your valuable and insightful comments. We address the questions and concerns below.
>
> >**W1**: The paper seems to heavily build upon pre-existing methods such as "graph-based reasoning systems" and "structured knowledge integration," which have been extensively explored in the literature (e.g.,logical NNs and other neuro-symbolic approaches)
>
> **Response for W1:**
>
> We thank the reviewer for their valuable feedback and for raising their concerns regarding the potential reliance on pre-existing methods in our proposed approach. We appreciate the opportunity to address this issue.
>
> We would like to emphasize the difference between our method and graph-based reasoning systems as well as neuro-symbolic approaches.
>
> ### Features of Graph-based reasoning systems and Neuro-symbolic approaches
> Graph-based reasoning systems typically represent knowledge and logical constraints explicitly as nodes and edges in a graph structure while neuro-symbolic approaches, Logic NNs for instance, generally translate logical operations such as AND, OR, and NOT into differentiable components that can be trained alongside neural network parameters. These methods perform reasoning relying on either the logical constraints contained in the graph structure or the trained logical  components.
>
> ### Difference with Graph-based reasoning systems and Neuro-symbolic approaches
> While the generation of concept maps and mind maps in our method may appear similar to graph-based reasoning systems, we demonstrate notable differences between them in three key aspects.
>
> Firstly, there are no logical operations involved during the generation of concept maps and mind maps. It is only a process to identify and organize relevant information. We leave all the reasoning part to LLMs.
>
> Secondly, the concept maps and mind maps are not kept once a more organized and concise form of context is obtained. We leave all the reasoning part to LLMs and the processed context is the only input to LLMs. The reasoning process is not constrained by any graph structure or logical  components.
>
> Thirdly, our method can be seen as a new prompt engineering technique without altering the underlying model architecture or training paradigms like Logic NNs do.
>
> Overall, by carefully adapting the given context to the inference process of LLMs in a concise and organized manner, we aim to lower the difficulty of proof planning for LLMs and unlock the deductive reasoning abilities that were previously untapped, while the generation of concept maps and mind maps in our method is just an approach to obtain a concise and organized context.
>
> >**W2**: The justification for why the combination of concise and organized strategies is more effective remains unclear under a rigorous theoretical framework. The examples provided (Sec 3 and Fig 3) do not offer evidence that this combination offers a qualitatively different approach to reasoning in LLMs as compared to existing methods.
>
> **Response for W2:**
>
> We would like to thank the reviewer for their valuable feedback and the opportunity to address their concern.
>
> ### Why the combination of concise and organized strategies is more effective
>
> Firstly, previous Literature [1] offers experimental evidence and states that the order of proofs affects reasoning.
>
> Secondly,  in this paper, we have conducted experiments in Figure 1 (d) to support our motivation. We randomly select 196 samples from the ProofWriter dataset and reconstruct the context based on the provided ground-truth proofs into either concise or organized forms while regarding the ground-truth proofs as concise and organized. The results in Figure 1(d) demonstrate that both organized and concise input context can greatly improve the reasoning accuracy of LLMs. The results also indicate the complementarity between concise and organized perception with the combination of them yields a relative performance improvement of over 100% (35.9% vs 71.9%) in a 5-hop setting. The reason why this combination can work might be that it greatly decreases the difficulty of proof planning which LLMs still struggle with.
>
> [1] Abulhair Saparov and He He. Language models are greedy reasoners: A systematic formal analysis of chain-of-thought. In The Eleventh International Conference on Learning Representations, 2023

---

> > ### Author Response · Authors · 2023-11-20
> > **Response to Reviewer fpHM [part 2/2]**
> >
> > >**W3**: The experimental results reported in Section 4.2 might not be robust enough. The benchmarks ProofWriter and ProtoQA, is not sufficiently demonstrating the general efficacy of COP across various datasets and reasoning tasks. Besides, the lack of a detailed error analysis or ablation study also keep the improvement of the whole system a mystery.
> >
> > **Response for W3:**
> >
> > Thanks for the helpful suggestion.
> >
> > As suggested, we further conduct experiments on FOLIO, a real-world logical reasoning benchmark with various type of rules.
> > To adapt to FOLIO, which is more complex and contains various language patterns and rule types, COP adopts a combination of rouge scores and semantic similarity method to generate the concept maps and mind maps. With the slight adjustment, COP outperforms CoT and LogicLM while LAMBADA is not able to work on this dataset, demonstrating the general efficacy of COP. The concise and organized context that COP provides on FOLIO facilitates the reasoning of LLMs while CoT still suffers from redundant and out-of-order context.
> >
> > | Methods  | Accuracy  |
> > |---|---|
> > |  Standard | 54.60  |
> > | CoT  | 57.84  |
> > | Logic-LM  | 61.76  |
> > | **COP**  | **65.27**  |
> >
> > ### Detailed error analysis
> >
> > The proposed method consists of several steps (i.e., Generation of Concept Maps, Generation of Mind Map, Context Reconstruction and Reasoning using LLMs). These steps are performed serially and it is not able to predict an answer without any of them. Therefore, we add the detailed error analysis as suggested. Notably, since context reconstruction is based on rules  and will not cause any errors, it is not included in the detailed error analysis.
> >
> > |   | **Generation of Concept Maps** | **Generation of Concept Maps** |  **Generation of Mind Map** | **Generation of Mind Map**  | **LLMs’ reasoning** |  **LLMs’ reasoning** | **LLMs’ reasoning**  |
> > |---|:-:|:-:|:-:|:-:|:-:|:-:|:-:|
> > |   |  Simplified Representation of Rules and Facts |  Connection of Rules and Facts |  Simplified Representation of Question  | Generation of mind map  | hallucination  |  Multi conditions |  Multi-hop planning |
> > |  Number of Error Cases |  8  |  13 | 0  | 3  | 10  | 12  | 20  |
> > |  Accuracy |  98.67% (592/600)  | 97.80% (579/592)  |  100% (579/576 /579) | 99.48% (576/579)  | 92.71% (534/576) | 92.71% (534/576) | 92.71% (534/576) |
> >
> > The error analysis is based on the results of d5 split on the ProofWriter dataset. The test set is made up of 600 samples. According to the table, the main failure of COP is attributed to the instability in the reasoning of LLMs even after all the previous steps are correct. We further divide these bad cases as three types: failure caused by hallucination, multi conditions and multi-hop planning. Examples of the three kinds are listed below:
> >
> > ### Example for hallucination:
> >
> > **The concise and organized context:** Anne is round. All round people are kind. Kind people are blue.Anne is round.If someone is round and blue then they are white.
> >
> > **The statement to determine whether it is True or False**: Anne is not white
> >
> > **Ground Truth:** False.
> >
> > **Proof steps produced by LLMs:** Anne is round. All round people are kind. Therefore Anne is kind. Kind people are blue. Therefore Anne is blue. If someone is round and blue then they are white. Therefore Anne is white. Therefore the given statement 'Anne is not white' is True
> >
> > Despite the model generate correct proof steps, it produce a wrong answer, which might be caused by hallucination.
> >
> > ### Example for failure caused by multi conditions:
> >
> > **The concise and organized context:** Harry is kind.All kind, red things are green.
> >
> > **The statement to determine whether it is True or False**: Harry is not green.
> >
> > **Ground Truth:** Unknown.
> >
> > **Proof steps produced by LLMs:** Harry is kind. All kind, red things are green. Therefore Harry is green. Therefore the given statement 'Harry is not green' is False.
> >
> > The model happens to wrongly process rules with multi conditions like “All kind, red things are green”.
> >
> > ### Example for failure caused by multi-hop planning:
> >
> > **The concise and organized context:** The cow chases the cat. If someone chases the cat then they visit the tiger. If someone visits the tiger then they visit the bald eagle.
> >
> > **The statement to determine whether it is True or False**: The cow does not visit the bald eagle.
> >
> > **Ground Truth:** False.
> >
> > **Proof steps produced by LLMs:** The cow chases the cat. If someone chases the cat then they visit the tiger. Therefore the given statement 'The cow does not visit the bald eagle' is Unknown.
> >
> > The model happens to failure of proof planning for multi-hop problems even if the context is concise and organized.
> >
> > The failure caused by multi-hop planning and the failure caused by hallucination can be further mitigated by decomposing the reasoning steps into smaller tasks like least-to-most prompting and strengthening of LLMs’ fundamental ability. We leave the failure caused by multi conditions as future work.

---

### Meta-Review · Area_Chair_Rz2c · 2023-12-13

**Metareview:**

**Paper Summary:**

This paper introduces Concise and Organized Perception (COP), a novel approach to enhance the deductive reasoning capabilities of large language models (LLMs). COP first creates mind maps to represent hierarchical relationships among facts and rules, followed by pruning to tailor the information flow for LLMs. The approach is tested on three synthetic logical reasoning datasets (ProofWriter, PrOntoQA, and PrOntoQA-OOD) and demonstrates improvements over existing methods.

**Strengths:**
1. Innovation in Deductive Reasoning: COP's combination of concise and organized strategies proposes a new way of improving LLMs' deductive reasoning (fpHM, Fc11).
2. Strong Results: The approach shows improvements in reasoning accuracy and efficiency while reducing the number of queries to LLMs (pmsh, Fc11, 3B4m).

**Weaknesses:**

1. Insufficient Experiments: The initial choice of datasets and lack of detailed error analysis raised questions about the general applicability of COP. However, the authors addressed this concern by including additional real-world dataset evaluations (fpHM, Fc11, 3B4m).
2. Theoretical Justification Lacking: The paper does not provide a strong theoretical framework to justify the effectiveness of combining concise and organized strategies (fpHM).
3. Presentation and Clarity Issues: Some reviewers noted difficulties in understanding the method (pmsh, 3B4m).

**Decision**

This paper was on the borderline, and the authors made extensive efforts to respond to reviewers concerns. However due to the issues raised above we felt the paper in its current state did not meet the ICLR acceptance threshold.

**Justification For Why Not Higher Score:**

Based on reviews, this paper lacks theoretical justifications and some reviewers also raised concerns about its clarity in presentation.

**Justification For Why Not Lower Score:**

The authors have made extra efforts in addressing reviewers' major concerns.

---

### Decision · Program_Chairs · 2024-01-16

Reject